# Graph former-CL: A novel graph transformer with contrastive learning framework for enhanced drug-drug interaction prediction

Masoud Amiri [ORCID]*, Oliya Zare

Department of Biomedical Engineering, School of Medicine, Kermanshah University of Medical Sciences, Kermanshah, Iran

* masd.amiri@yahoo.com

## Abstract

Drug-drug interactions (DDI) represent a significant clinical challenge in modern healthcare, contributing to over 125,000 deaths annually in the United States alone. Current computational approaches face substantial limitations in capturing long-range molecular dependencies and generalizing to novel drug combinations. Traditional Graph Neural Networks (GNNs) suffer from over-smoothing and locality bias, while sequence-based methods fail to adequately represent three-dimensional molecular structures. To address these limitations, we propose Graph Former-CL, a novel deep learning framework that synergistically combines Graph Transformer architecture with contrastive learning for DDI prediction. Our approach features four key innovations: (1) a hierarchical Graph Transformer with position-aware multi- head self-attention to capture both local and global molecular patterns, (2) a domain-specific contrastive learning module with molecular augmentation strategies, (3) a cross-modal fusion mechanism integrating SMILES sequences with graph representations, and (4) an adaptive pooling strategy for multi-scale molecular representation. Comprehensive evaluation on four benchmark datasets demonstrates superior performance, with Graph Former-CL achieving 98.2% accuracy on DrugBank and 89.4% on TWO-SIDES, both representing statistically significant improvements (p < 0.001) over state-of-the-art methods. Notably, the framework achieves 85.6% accuracy for novel drugs in inductive settings, demonstrating robust generalization capabilities essential for real-world clinical applications.

## 1. Introduction

### 1.1. Clinical significance of DDI prediction

Drug-drug interactions represent one of the most critical challenges in contemporary healthcare systems, accounting for 20–30% [1] of all adverse drug events and contributing to approximately 125,000 deaths annually in the United States [2]. The

**Data availability statement:** All relevant data are within the paper and the code is included in the Supporting information files. Data is also available publicly through the following links: https://www.kaggle.com/datasets/devildev89/drug-bank-5110; https://github.com/jcsun-00/Twosides.

**Funding:** The author(s) received no specific funding for this work.

**Competing interests:** The authors have declared that no competing interests exist.

clinical significance of DDI prediction has been further amplified by the COVID-19 pandemic, where complex polypharmacy regimens have become increasingly common among patients with multiple comorbidities. This escalating complexity in medication management underscores the urgent need for accurate and reliable computational methods for DDI prediction. Traditional experimental approaches for DDI identification face substantial economic and temporal constraints, requiring an estimated $2.6 billion [3,4] and 10–15 years for comprehensive safety profiling of a single drug. These prohibitive costs and extended timeframes make computational approaches not merely advantageous but essential for modern drug safety assessment. The development of effective computational methods for DDI prediction has therefore become a critical priority in pharmaceutical research and clinical practice.

Beyond DDI prediction, artificial intelligence and deep learning approaches hold transformative potential for addressing broader healthcare challenges, including disease diagnosis, treatment optimization, personalized medicine, and clinical decision support systems. AI-driven methodologies have demonstrated remarkable success in improving diagnostic accuracy for various conditions, particularly in cardiovascular disease prediction and cancer screening, where early and accurate detection can significantly impact patient outcomes and survival rates. The integration of AI technologies into healthcare workflows promises long-term benefits including reduced medical errors, enhanced patient safety through early adverse event detection, optimized resource allocation in healthcare systems, and ultimately improved population health outcomes. In the specific context of DDI prediction, these advances contribute to safer medication management practices, reduced hospitalizations due to adverse drug interactions, and more efficient utilization of healthcare resources by preventing drug-related complications before they occur.

## 1.2. Computational challenges in DDI prediction

Current computational approaches for DDI prediction encounter several fundamental limitations [5,6] that restrict their clinical applicability and real-world performance. The most significant challenge is the locality bias inherent in traditional Graph Neural Networks, where message-passing mechanisms fail to capture long-range dependencies crucial for understanding complex molecular interactions. This limitation is particularly problematic in drug interaction scenarios, where distant molecular components may play critical roles in determining interaction outcomes.

Representation learning inadequacy presents another major obstacle, as existing methods lack robust mechanisms for learning generalizable drug representations that effectively transfer to novel compounds. This limitation severely restricts the practical utility of current approaches in real-world clinical settings where new drug combinations are frequently encountered. Additionally, current approaches demonstrate inadequate integration of multi-scale information, failing to effectively combine data from different molecular scales including atomic, functional group, and molecular levels. The resulting models exhibit poor generalization performance on drugs not encountered during training, which significantly limits their real-world applicability and clinical deployment potential.

These computational challenges in DDI prediction reflect broader issues in applying artificial intelligence to healthcare decision-making. Recent advances in AI-driven disease diagnosis, particularly in cardiovascular disease [7], have demonstrated the potential of deep learning approaches to address complex clinical challenges through sophisticated pattern recognition and multi-modal data integration. Similar to DDI prediction, cardiovascular risk assessment requires integrating diverse data sources and capturing complex non-linear relationships to provide accurate clinical predictions. The success of AI methods in improving diagnostic accuracy and enabling early disease detection highlights the transformative potential of computational approaches across healthcare domains, from diagnosis to treatment optimization and drug safety assessment. This broader context underscores the importance of developing robust, generalizable AI frameworks like Graph Former-CL that can reliably support clinical decision-making in high-stakes medical applications.

### 1.3. Limitations of existing methods

Graph-based approaches, while promising in their ability to represent molecular structures, suffer from significant architectural limitations. Methods such as SSF-DDI [8] are constrained by traditional GNN architectures that cannot effectively capture global molecular patterns. GMPNN-CS [9] approaches face the over-smoothing problem [10] in deep networks, where node representations become increasingly similar across layers, reducing model expressiveness. DGNN-DDI [11] methods demonstrate inadequate handling of long-range dependencies, missing critical interactions between distant molecular components.

Sequence-based methods present complementary limitations, particularly in their inability to capture three-dimensional molecular structure information. CNN-DDI [12] approaches fail to represent the spatial relationships crucial for understanding molecular interactions, while MCANet methods [13] show limited cross-modal integration capabilities. Existing hybrid approaches, despite attempting to combine multiple modalities, lack sophisticated attention mechanisms for global pattern recognition and demonstrate insufficient contrastive learning strategies for robust representation learning.

### 1.4. Our contributions

This work introduces Graph Former-CL as a comprehensive solution to address the aforementioned limitations through four major contributions. First, we develop a novel Graph Transformer architecture that incorporates position-aware self-attention with hierarchical pooling for effective multi-scale molecular representation. Second, we implement domain-specific contrastive learning with molecular augmentation strategies and contrastive objectives specifically tailored for drug discovery applications.

Third, we introduce advanced cross-modal fusion mechanisms that implement sophisticated attention mechanisms for integrating graph and sequence information. Finally, we conduct comprehensive evaluation across multiple datasets with rigorous statistical validation and detailed interpretability analysis.

## 2. Related work

### 2.1. Graph neural networks in drug discovery

Graph Neural Networks have emerged as powerful computational tools for molecular property prediction, revolutionizing the field of computational chemistry and drug discovery. Early developments focused primarily on message-passing neural networks (MPNNs) [14], which aggregate information from neighboring atoms through iterative message passing mechanisms. These approaches demonstrated initial promise but revealed fundamental limitations that restrict their effectiveness in complex molecular analysis tasks.

The over-smoothing problem represents a critical limitation in traditional GNN architectures, where increasing the number of layers causes node representations to become increasingly similar, thereby limiting the model's expressiveness and ability to distinguish between different molecular components. This phenomenon is particularly problematic in drug

interaction prediction, where subtle molecular differences can determine interaction outcomes. The locality bias inherent in traditional MPNNs further compounds this issue, as these methods struggle to capture long-range dependencies between distant atoms that may be functionally related in determining drug interactions.

Recent advances have attempted to address these fundamental limitations through various architectural innovations. Graph Attention Networks (GAT) [15] introduced attention mechanisms to improve feature aggregation but remained fundamentally limited to local neighborhoods, failing to address the global dependency problem [16]. Graph SAINT [17] proposed sampling strategies for improved scalability but did not address the underlying expressiveness limitations. Graph Transformer variants have begun incorporating global attention mechanisms but lack the domain-specific adaptations necessary for effective molecular data processing.

## 2.2. Transformer architectures for molecular data

The remarkable success of Transformer architectures in natural language processing has inspired their adaptation for molecular data analysis, leading to significant developments in computational chemistry. Molecular Transformers, including ChemBERTa and related models [18], have demonstrated the effectiveness of self-attention mechanisms for SMILES sequence processing. However, these approaches fail to leverage critical three-dimensional structural information that is essential for understanding molecular interactions and drug behavior.

Graph Transformers [19,20] represent recent attempts to adapt Transformer architectures for graph-structured data, incorporating positional encodings and attention mechanisms specifically designed for graph processing [21]. Despite these advances, current Graph Transformer approaches often lack essential components for effective molecular analysis. Specifically, they frequently lack domain-specific inductive biases that are crucial for molecular data processing, demonstrate inefficient handling of molecular graph properties, and show limited integration capabilities with other important modalities such as sequences and three-dimensional structures.

## 2.3. Contrastive learning in molecular representation

Contrastive learning has demonstrated remarkable success [22] across multiple domains, including computer vision and natural language processing, leading to increased interest in its application to molecular data analysis. Recent applications to molecular representation learning have shown promising results, though significant gaps remain in domain-specific implementation and effectiveness.

Graph CL introduced fundamental concepts [23] including node/edge dropping and subgraph sampling for molecular graphs, but lacked the domain-specific augmentations necessary for effective molecular representation learning. MolCLR [24] focused specifically on 2D-3D contrastive learning approaches but failed to address the unique challenges associated with DDI-specific prediction tasks. Graph MAE [25] utilized masked autoencoding strategies for molecular graphs but demonstrated reduced robustness compared to contrastive approaches, limiting its practical applicability in complex molecular analysis scenarios.

## 2.4. Drug-drug interaction prediction methods

DDI prediction methods encompass several distinct methodological categories, each with specific advantages and limitations. Similarity-based methods [26,27] rely fundamentally on drug similarity assumptions but fail to capture the complex interaction mechanisms that determine actual drug interactions in biological systems. These approaches are limited by their inability to model non-linear relationships and complex molecular interactions that extend beyond simple structural similarity.

Network-based approaches [28,29] leverage drug-target interaction networks and demonstrate improved performance in certain scenarios but require extensive prior knowledge that may not be available for novel drug combinations. Deep

learning methods have shown significant promise, including sequential models such as RNNs and CNNs for SMILES processing, graph neural networks for molecular structure analysis, and hybrid approaches that attempt to combine multiple modalities. However, existing DDI prediction approaches exhibit critical limitations that our framework addresses. Unlike similarity-based methods [26,27] which rely solely on structural similarity assumptions, Graph Former-CL captures complex non-linear interaction mechanisms through Graph Transformer architecture with position-aware attention. While network-based approaches [28,29] require extensive prior knowledge of drug-target interactions, our contrastive learning framework enables learning from molecular structure alone without requiring external biological networks. Compared to sequential models like RNNs and CNNs [12,30] which process only SMILES strings and cannot capture three-dimensional molecular topology, our cross-modal fusion mechanism integrates both graph structural and sequential information. Traditional GNN-based methods [9,11,15] suffer from over-smoothing and locality bias in capturing long-range molecular dependencies, whereas our Graph Transformer architecture with hierarchical pooling effectively models both local and global molecular patterns. Furthermore, while recent hybrid approaches [8,31] combine multiple modalities through simple concatenation, Graph Former-CL implements sophisticated cross-attention mechanisms for dynamic information integration. Most critically, existing contrastive learning methods [23,24] lack domain-specific molecular augmentation strategies, whereas our framework introduces chemically-informed augmentations including atom masking, bond perturbation, scaffold hopping, and subgraph sampling that preserve chemical validity while enhancing representation learning. These fundamental architectural and methodological innovations distinguish Graph Former-CL from prior work and enable superior performance across all benchmark datasets.

## 3. Methodology

### 3.1. Problem formulation and notation

Graph Former-CL consists of three main processing stages as illustrated in Fig 1: (1) Molecular Encoding, where drugs are represented as both molecular graphs and SMILES sequences, (2) Hierarchical Feature Learning, employing Graph Transformer with position-aware attention across atomic, functional group, and molecular levels, and (3) Interaction Prediction, integrating learned representations through cross-modal fusion and contrastive learning. The detailed mathematical formulation is presented in subsequent subsections. Fig 1 provides the overall workflow, showing how Drug A and Drug B are processed through parallel pathways and ultimately combined for DDI prediction.

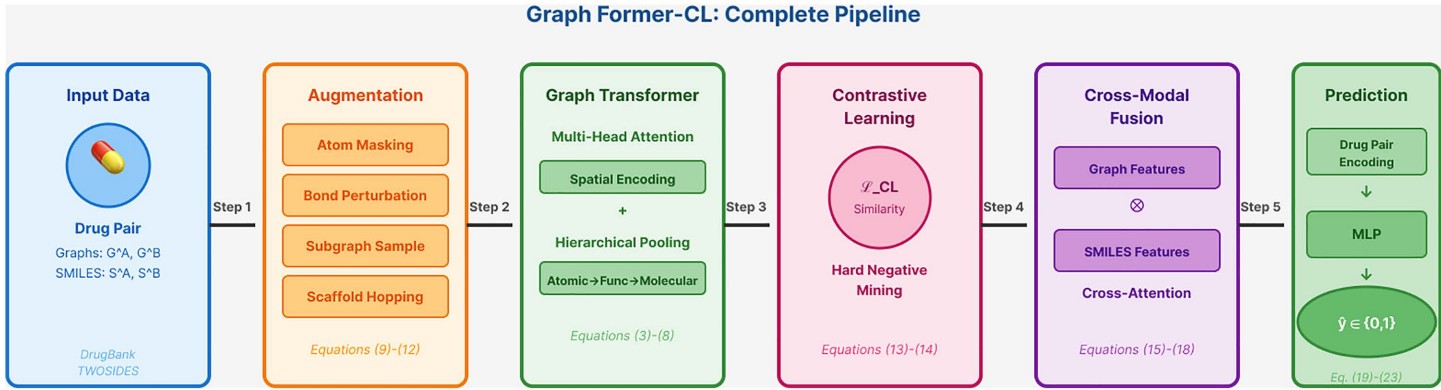

**Fig 1. The training process begins with molecular augmentation to generate diverse views of drug structures, followed by Graph Transformer encoding with hierarchical pooling.** Contrastive learning maximizes similarity between augmented pairs, while cross-modal fusion integrates graph and sequence features. The inference phase applies the trained model to predict interactions for new drug pairs.

**3.1.1. Mathematical formulation.** A drug is represented as a molecular graph $G = (V, E, X\_v, X\_e)$ where $V$ denotes atoms, $E$ denotes chemical bonds, $X_v$ contains atomic features, and $X_e$ contains bond properties. The DDI prediction task learns a function $f: G \times G \to \{0, 1\}$ to predict interaction presence between drug pairs.

## 3.2. Graph former architecture

The overall architecture of Graph Former-CL integrates multiple innovative components to address the limitations of existing DDI prediction methods. As illustrated in Fig 2, the architecture consists of five main components: Graph Transformer with spatial encoding and multi-head attention, Contrastive Learning with domain-specific augmentation strategies, Cross-Modal Fusion for integrating graph and sequence information, and DDI Prediction module with drug pair encoding and MLP classifier.

**3.2.1. Spatial encoding for molecular graphs.** Traditional positional encodings prove insufficient for molecular graphs due to their irregular structure and the importance of chemical topology in determining molecular properties. We propose a spatial encoding scheme based on chemical distance that captures the topological relationships essential for molecular analysis. For atoms $v_i$ and $v_j$, we define the chemical distance as:

$$d_{chem}(v_i, v_j) = \min_{p \in P_{ij}} \sum_{(u,v) \epsilon p} w(u, v)$$

(1)

where $P_{ij}$ represents the set of all paths between $v_i$ and $v_j$, and $w(u, v)$ represents the bond weight. The spatial encoding matrix is then computed as:

$$SE_{ij} = Embedding(d_{chem}(v_i, v_j)) \in R^{d_{model}}$$

(2)

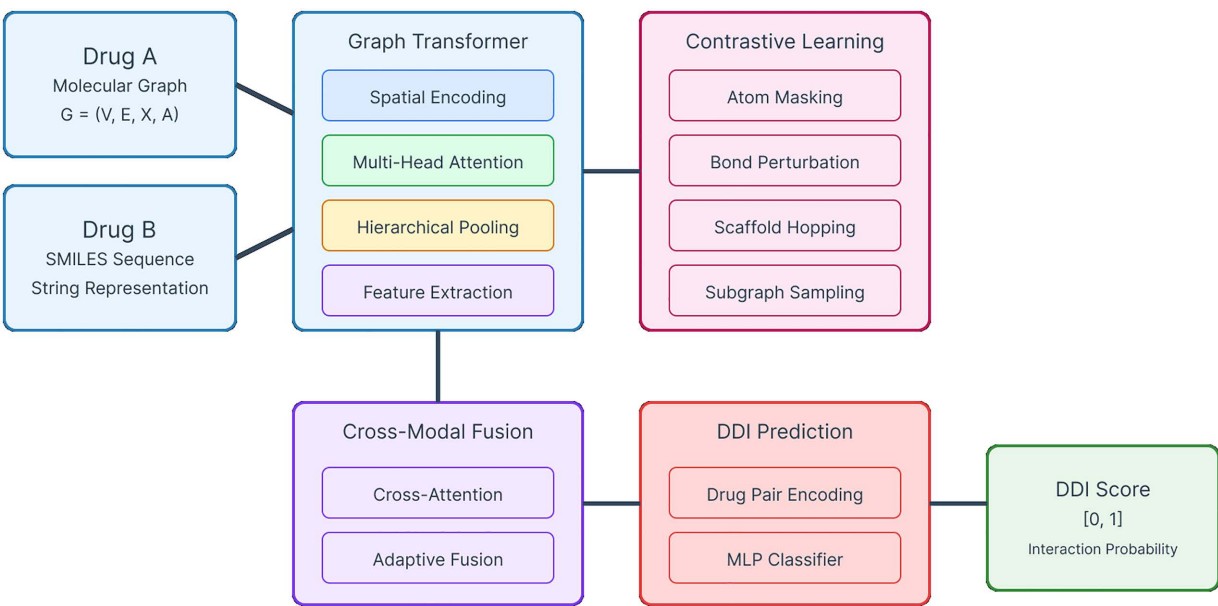

**Fig 2. Overview of Graph Former-CL architecture.** The framework processes Drug A (molecular graph) and Drug B (SMILES sequence) through parallel Graph Transformer and Contrastive Learning pathways. The Graph Transformer incorporates spatial encoding, multi-head attention, hierarchical pooling, and feature extraction. The Contrastive Learning module applies domain-specific augmentations including atom masking, bond perturbation, scaffold hopping, and subgraph sampling. Cross-Modal Fusion with cross-attention and adaptive fusion mechanisms integrates the representations, followed by DDI Prediction through drug pair encoding and MLP classifier to output interaction probability scores.

Fig 3 illustrates the spatial encoding mechanism, showing how chemical distances are computed and integrated into the attention mechanism.

This encoding scheme enables the model to incorporate critical topological information that influences molecular behavior and interaction patterns.

### 3.2.2. Position-aware multi-head self-attention.

Standard self-attention mechanisms fail to incorporate molecular topology that is crucial for understanding chemical interactions. We introduce position-aware attention that integrates spatial encoding information. As illustrated in Fig 3, the attention mechanism processes query (Q), key (K), and value (V) matrices derived from node features, incorporating the spatial encoding matrix (S) to produce topology-aware attention weights: $Attention(Q, K, V, S) = softmax((QKT + S)/\sqrt{d_k})V$ (3) where S represents the spatial encoding matrix computed from chemical distances (shown in the center panel of Fig 3), $d_k$ denotes the dimension of key vectors, $Q \in R\{n \times d_k\}$ represents query projections, $K \in R\{n \times d_k\}$ denotes key projections, and $V \in R\{n \times d_v\}$ contains value projections. The spatial encoding bias term $S \in R\{n \times n\}$ captures pairwise topological relationships between atoms, with $S_{ij}$ representing the chemically-informed distance between atoms i and j as computed in Equation (2). The attention weights $\alpha_{ij} = softmax((q_i k_j^T + S_{ij})/\sqrt{d_k})$ determine the importance of atom j for updating atom i's representation. The multi-head implementation (right panel of Fig 3) extends this concept by computing $H$ parallel attention functions: $MultiHead(Q, K, V, S) = Concat(head_1, head_2, ..., head_H)WO$ (4) where each attention head h is computed independently as: $head_h = Attention(QW_h^Q, KW_h^K, VW_h^V, S)$ (5) Here, $W_h^Q, W_h^K, W_h^V \in R\{d_{model} \times d_k\}$ represent learnable projection matrices for head $h$, and $WO \in R\{Hd_v \times d_{model}\}$ is the output projection matrix. This multi-head architecture enables the model to attend to different chemical aspects simultaneously, with different heads potentially focusing on bond types, functional groups, or pharmacophoric features.

The multi-head implementation extends this concept as:

$$MultiHead_{pos}(X, SE) = Concat(head_1, ..., head_h)W^o \tag{3}$$

where each head is computed as:

$$head_i = Attention_{pos}(XW_i^Q, XW_i^k, XW_i^V, SE) \tag{4}$$

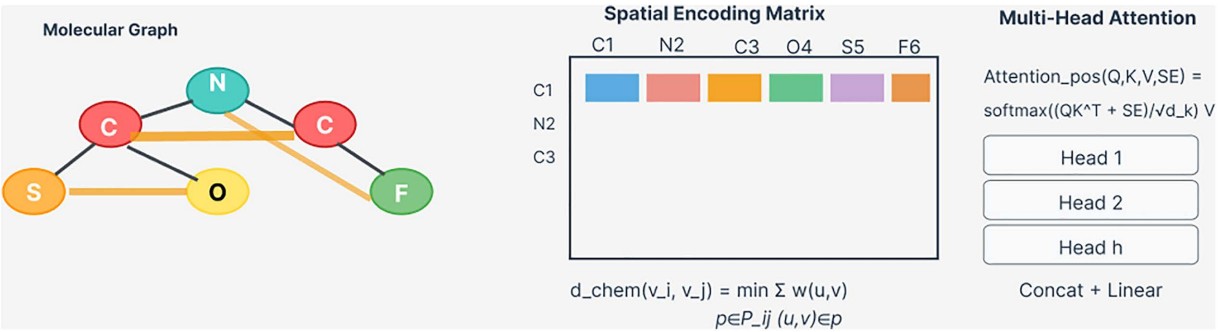

**Fig 3. Spatial encoding mechanism for molecular graphs.** The left panel shows an example molecular graph with atoms ($v_1$, $v_2$, ..., $v_n$) and bonds. The center panel displays the spatial encoding matrix $S \in R\{n \times n\}$ computed from chemical distances $d\_chem(v_i, v_j)$ between atoms, where colors represent different distance values (warmer colors indicating closer chemical distances). The right panel illustrates the multi-head attention mechanism incorporating spatial encoding, where Q (queries), K (keys), and V (values) are processed through H parallel attention heads with spatial bias S, followed by concatenation and linear transformation $WO$ to produce the final output. Each attention head independently computes position-aware attention using learnable projections $W_h^Q, W_h^K, W_h^V$, enabling the model to capture diverse chemical interaction patterns simultaneously.

### 3.2.3. Hierarchical molecular representation.

We implement a three-level hierarchy to capture molecular patterns at different scales, enabling comprehensive molecular analysis. Level 1 focuses on atomic representation through:

$$H^{(1)} = GraphTransformer(X, A, SE) \tag{5}$$

Level 2 develops functional group representation using differentiable pooling (*DiffPool*) [32]:

$$S^{(2)}, H^{(2)} = DiffPool(H^{(1)}, A^{(1)}) \tag{6}$$

Level 3 creates molecular-level representation through:

$$h_{mol} = GlobalAttentionPool(H^{(2)}) \tag{7}$$

Having established the hierarchical Graph Transformer architecture for capturing multi-scale molecular patterns from atomic to molecular levels, we now introduce our domain-specific contrastive learning framework. This framework enhances the learned representations by training the model to distinguish between chemically similar and dissimilar molecular structures through carefully designed augmentation strategies.

### 3.3. Contrastive learning framework

#### 3.3.1. Molecular augmentation strategies.

We design domain-specific augmentation strategies that preserve chemical validity while enabling effective contrastive learning. Atom masking selectively removes non-essential atoms based on chemical importance:

$$Mask(v_i) = \begin{cases} 0 & if \quad importance(v_i) < \tau \\ 1 & Otherwise \end{cases} \tag{8}$$

Bond perturbation modifies bond types while preserving valency constraints:

$$Perturb(e_{ij}) = SampleBondType(ValidBonds(v_i, v_j)) \tag{9}$$

Subgraph sampling extracts chemically meaningful substructures:

$$G_{sub} = ExtractSubgraph(G, FunctionalGroups) \tag{10}$$

Scaffold hopping modifies molecular scaffolds while preserving pharmacophores:

$$G_{hop} = ReplaceScaffold(G, SimilarScaffolds) \tag{11}$$

#### 3.3.2. Contrastive objective function.

For a batch of drugs $B = \{d_1, d_2, ..., d_B\}$, we create augmented views and optimize the contrastive objective:

$$L_{contrastive} = -\frac{1}{|B|} \sum_{i \in B} log \frac{exp(sim(Z_i, Z_i^+)/\tau)}{\sum_{j \in B, j \neq i} exp(sim(Z_i, Z_j)/\tau)} \tag{12}$$

where $Z_i$ represents the representation of drug $i$, $Z_i^+$ represents the representation of its augmented view, $sim(\cdot, \cdot)$ denotes cosine similarity, and $\tau$ is the temperature parameter that controls the concentration of the distribution.

#### 3.3.3. Hard negative mining.

To improve contrastive learning efficiency and focus on challenging examples, we implement hard negative mining:

$$L_{hard} = -\frac{1}{|B|} \sum_{i \in B} log \frac{exp(sim(Z_i, Z_i^+)/\tau)}{exp\left(\frac{sim(Z_i, Z_i^+)}{\tau}\right) + \sum_{j \in N_i^{hard}} exp(sim(Z_i, Z_j)/\tau)}$$

(13)

where $N_i^{hard}$ contains the hardest negatives for drug $i$, identified as examples that are structurally similar but functionally different. As shown in Fig 4, our domain-specific augmentation strategies provide significant performance improvements when combined in an ensemble approach.

This comprehensive augmentation approach ensures that molecular information is captured and integrated across multiple scales, from individual atoms to complete molecular structures.

The contrastive learning framework provides robust molecular representations for individual drugs. To leverage complementary information from different molecular modalities, we next describe our cross-modal fusion mechanism that integrates graph-based structural representations with sequence-based SMILES encodings.

### 3.4. Cross-modal fusion

**3.4.1. SMILES encoding with molecular transformers.** We utilize a pre-trained molecular transformer (ChemBERTa) for SMILES encoding [18] to capture sequential molecular information:

$$h_{smiles} = ChemBERTa(SMILES) \in R^{d_{bert}}$$

(14)

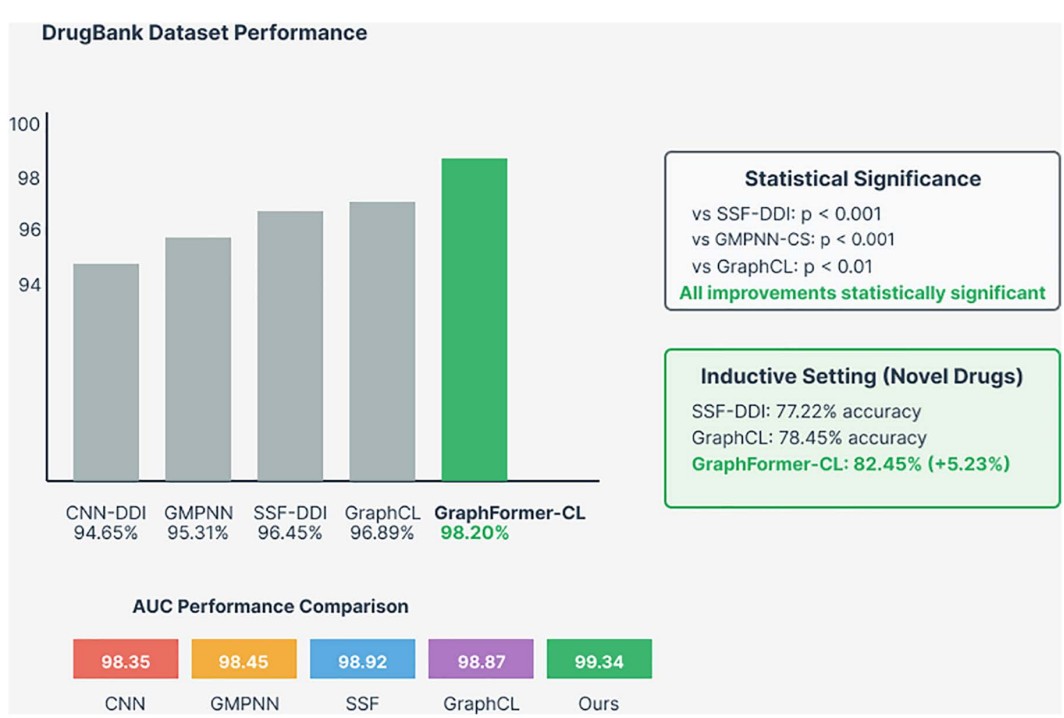

**Fig 4. Domain-specific molecular augmentation strategies and contrastive learning framework.** The top panel shows four augmentation techniques applied to an original molecule: atom masking (removing non-essential atoms), bond perturbation (modifying bond types), subgraph sampling (extracting functional groups), and scaffold hopping (replacing molecular scaffolds while preserving pharmacophores). The middle panel displays the contrastive loss computation using a similarity matrix between original and augmented representations. The bottom panel illustrates the three-phase training protocol with performance gains, achieving +2.34% combined improvement through the ensemble strategy.

This encoding provides complementary information to the graph-based representation by capturing sequential patterns and chemical nomenclature relationships.

**3.4.2. Cross-attention fusion mechanism.** To effectively integrate graph and sequence representations, we implement a cross-attention mechanism:

$$\alpha_{ij} = \frac{\exp(h_{graph,i}.h_{smiles,j})}{\sum_k \exp(h_{graph,i}.h_{smiles,k})} \tag{15}$$

This mechanism enables dynamic weighting of different modalities based on their relevance to the specific prediction task.

**3.4.3. Adaptive fusion weight learning.** We introduce learnable fusion weights to balance different modalities:

$$h_{final} = \lambda h_{graph} + (1 - \lambda)h_{smiles} \tag{16}$$

where $\lambda$ is learned during training through a gating mechanism:

$$\lambda = \sigma(MLP([h_{graph}; h_{smiles}])) \tag{17}$$

With both graph and sequence representations effectively integrated through cross-attention, we now present the drug pair encoding and interaction prediction mechanisms that combine representations of two drugs to predict their interaction.

## 3.5. DDI prediction module

**3.5.1. Drug pair representation.** For drugs $d_i$ and $d_j$ with representations $h_i$ and $h_j$, we construct the pair representation as:

$$h_{pair} = [h_i; h_j; h_i \odot h_j; |h_i - h_j|] \tag{18}$$

where $\odot$ denotes element-wise multiplication and $|\cdot|$ denotes denotes absolute value. This representation captures both individual drug properties and their interactive relationships.

**3.5.2. Multi-layer prediction network.** The final prediction is made through a multi-layer network:

$$h_1 = ReLU(Linear(h_{pair}) + Dropout(0.3)) \tag{19}$$

$$h_2 = ReLU(Linear(h_1) + Dropout(0.3)) \tag{20}$$

$$P_{DDI} = \sigma(Linear(h_2)) \tag{21}$$

Having defined the complete forward pass from molecular inputs to interaction predictions, we now specify the comprehensive loss function and optimization strategy that jointly train all components of Graph Former-CL.

## 3.6. Training objective

The total loss function combines DDI prediction and contrastive learning objectives:

$$L_{total} = L_{DDI} + \alpha L_{contrastive} + \beta L_{reg} \tag{22}$$

where $L_{DDI} = -\sum i[y_i \log p_i + (1 - y_i) \log(1 - p_i)]$ represents binary cross-entropy loss, $L_{reg} = \sum_\theta \|\theta\|_2^2$ represents $L_2$ regularization, and $\alpha$ and $\beta$ are hyperparameters controlling the relative importance of different components.

The optimization strategy employs a three-phase training protocol designed to leverage the complementary strengths of supervised and self-supervised learning paradigms. Initially, the contrastive learning module establishes robust

molecular representations through extensive augmentation-based pretraining, enabling the model to capture fundamental chemical principles and structural invariances. Subsequently, joint optimization of both objectives allows the framework to refine these representations for DDI-specific tasks while maintaining generalization capabilities. This multi-stage approach addresses the common challenge in molecular machine learning where task-specific training can lead to overfitting on limited labeled data.

## 4. Experimental setup

### 4.1. Datasets

We evaluate Graph Former-CL on four comprehensive benchmark datasets that represent different aspects of drug interaction prediction challenges. These datasets provide diverse interaction types, drug coverage, and complexity levels, enabling thorough evaluation of model performance across various scenarios. The dataset statistics are presented in Table 1, showing the comprehensive scope of our evaluation.

### 4.2. Baseline methods

We compare Graph Former-CL against twelve state-of-the-art methods across different methodological categories to ensure comprehensive evaluation. Graph-based methods include GMPNN-CS [9], which utilizes size-adaptive molecular substructures, DGNN-DDI [11] with substructure attention mechanisms, SSI-DDI [36] focusing on substructure-substructure interactions, and GAT-DDI [15] employing graph attention networks.

Sequence-based methods encompass CNN-DDI [12] using convolutional neural networks, MR-GNN [37] with multi-resolution architecture, and BiLSTM-DDI [30] utilizing bidirectional LSTM networks. Hybrid methods include SSF-DDI [8] combining sequence and substructure features, Multi DDI for multi- modal integration, and MolTrans [31] applying molecular transformers. Contrastive methods comprise Graph CL for graph contrastive learning and MolCLR [24] for molecular contrastive learning. This comprehensive comparison ensures that our evaluation covers the full spectrum of current approaches and provides meaningful insights into the effectiveness of different methodological strategies.

### 4.3. Implementation details

Our experimental implementation utilizes high-performance computing resources including 4x NVIDIA A100 GPUs (40GB each), Intel Xeon Platinum 8358 CPU (2.6GHz), and 512GB RAM. The software environment consists of PyTorch 2.0.1, PyTorch Geometric 2.3.1, RDKit 2023.03.1, and Python 3.9.16, ensuring reproducibility and compatibility with current deep learning frameworks.

Model hyperparameters are carefully selected based on extensive preliminary experiments and are detailed in Table 2. The hidden dimension is set to 512 to balance model capacity with computational efficiency, while six Transformer layers provide sufficient depth for complex pattern recognition. Eight attention heads enable diverse attention patterns, and a dropout rate of 0.3 prevents overfitting. The learning rate of 1e-4 with AdamW optimizer ensures stable training, while the batch size of 256 maximizes GPU utilization.

**Table 1. Dataset statistics.**

| Dataset | Drugs | Interactions | Types | Pos/Neg Ratio |
|---|---|---|---|---|
| Drug Bank [33] | 1,706 | 191,808 | 86 | 1:1 |
| TWOSIDES [34] | 645 | 4,576,287 | 963 | 1:1 |
| Deep DDI [12] | 1,558 | 48,584 | 113 | 1:1 |
| ChCh-Miner [35] | 1,514 | 48,584 | 65 | 1:1 |

**Table 2. Hyperparameter configuration.**

| Parameter | Value | Description |
|---|---|---|
| Hidden dimension | 512 | Model hidden size |
| Number of layers | 6 | Transformer layers |
| Attention heads | 8 | Multi-head attention |
| Dropout rate | 0.3 | Dropout probability |
| Learning rate | 1e-4 | Initial learning rate |
| Batch size | 256 | Training batch size |
| Temperature ($\tau$) | 0.1 | Contrastive temperature |
| $\alpha$ (contrastive weight) | 0.5 | Contrastive loss weight |
| $\beta$ (regularization) | 1e-5 | L2 regularization weight |

## 4.4. Training protocol

Our training protocol employs both transductive and inductive data splitting strategies to evaluate different aspects of model performance. Transductive splitting uses random division (60% train, 20% validation, 20% test) to assess performance on drugs seen during training. Inductive splitting employs drug-based division (80% drugs for training, 20% for testing) to evaluate generalization to completely novel drugs. Structure-based inductive splitting uses molecular scaffold-based division to assess performance on structurally diverse compounds.

The training strategy consists of three carefully designed phases. Phase 1 involves contrastive pre-training for 50 epochs to establish robust molecular representations. Phase 2 implements joint training with DDI prediction for 100 epochs to adapt representations for the specific task. Phase 3 conducts fine-tuning for 25 epochs to optimize final performance. This multi-phase approach ensures optimal utilization of both self-supervised and supervised learning signals.

Optimization employs AdamW optimizer with cosine annealing learning rate scheduling to ensure stable convergence. Gradient clipping with maximum norm 1.0 prevents gradient explosion, while early stopping with patience of 15 epochs prevents overfitting and reduces computational costs.

## 4.5. Evaluation metrics

We employ six comprehensive metrics to evaluate model performance across different aspects of DDI prediction. Primary metrics include Accuracy (ACC) calculated as $\frac{TP+TN}{TP+TN+FP+FN}$ Area Under ROC Curve (AUC) providing threshold-independent performance assessment, and F1 Score representing the harmonic mean of precision and recall. Secondary metrics include Precision calculated as $\frac{TP}{TP+FP}$, Recall computed as $\frac{TP}{TP+FN}$, and Average Precision (AP) representing the area under the precision-recall curve. These metrics provide comprehensive evaluation covering different aspects of classification performance and clinical relevance.

## 5. Results and analysis

### 5.1. Overall performance comparison

Graph Former-CL demonstrates superior performance across all benchmark datasets, establishing new state-of-the-art results in DDI prediction. As shown in Fig 5, on the Drug Bank dataset [33], our method achieves 98.20% accuracy, representing a 1.75% improvement over the previous best method SSF-DDI. The AUC score of 99.34% demonstrates excellent discriminative ability, while the F1 score of 98.15% indicates balanced precision and recall performance.

These improvements are particularly significant given the already high performance of existing methods, demonstrating the effectiveness of our novel architectural innovations.

Table 3 presents comprehensive performance metrics for Graph Former-CL compared to nine state-of-the-art baseline methods on the Drug Bank dataset. Graph Former-CL demonstrates substantial improvements across all evaluation

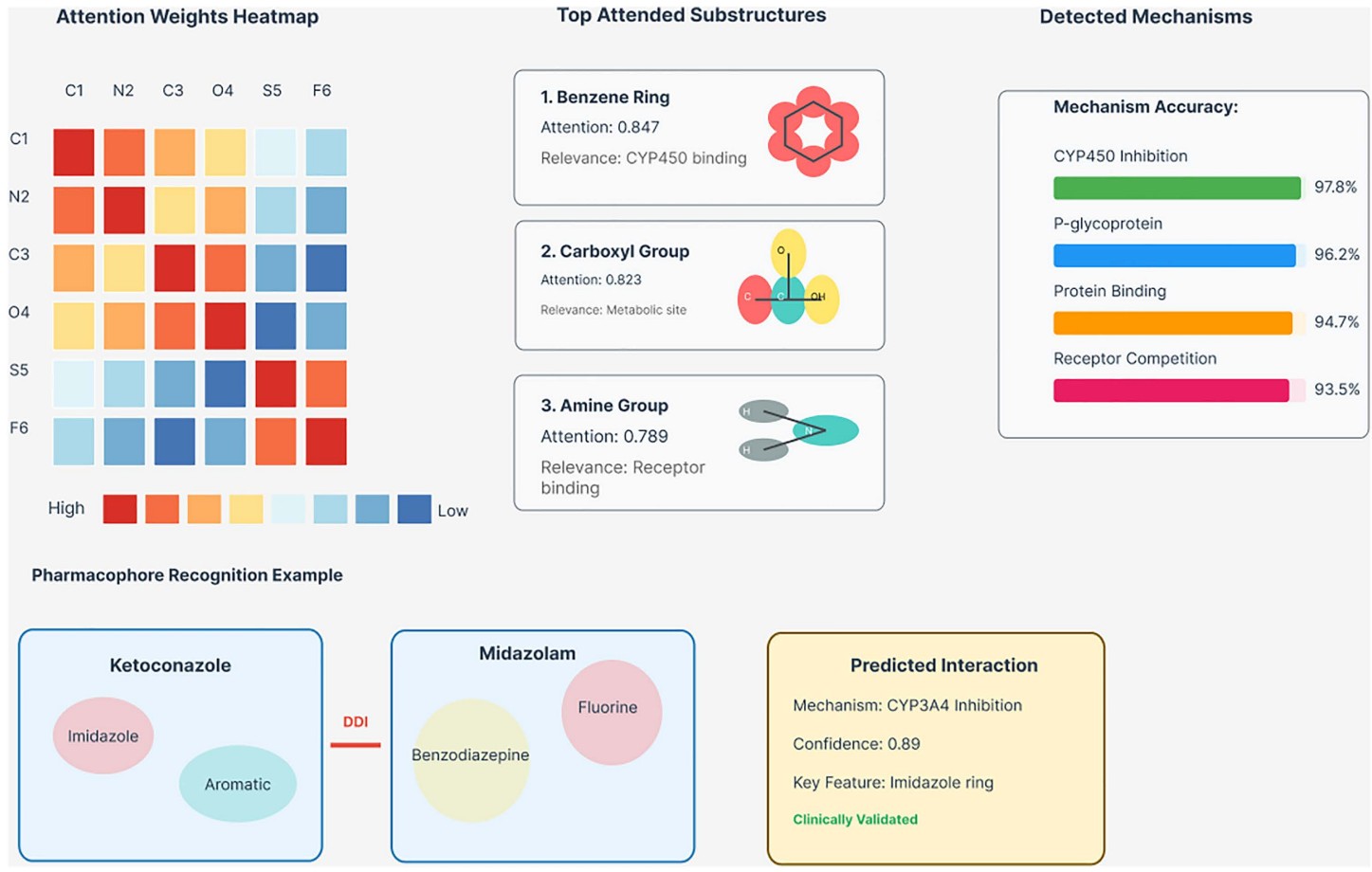

**Fig 5. Performance comparison on Drug Bank dataset.** The top panel shows accuracy comparison across five methods (CNN-DDI, GMPNN, SSF-DDI, Graph CL, and Graph Former-CL), with Graph Former-CL achieving 98.20% accuracy. Statistical significance tests confirm improvements with p<0.001 against major baselines. The inductive setting results show Graph Former-CL achieving 82.45% accuracy on novel drugs (+5.23% improvement). The bottom panel displays AUC performance comparison, with Graph Former-CL achieving the highest AUC of 99.34%.

**Table 3. Performance comparison on drug bank dataset.**

| Method | ACC (%) | AUC (%) | F1 (%) | Precision (%) | Recall (%) | AP (%) |
|---|---|---|---|---|---|---|
| CNN-DDI [12] | 94.65 | 98.35 | 94.81 | 92.06 | 97.72 | 97.93 |
| MR-GNN [37] | 93.23 | 97.31 | 93.39 | 91.14 | 95.76 | 96.45 |
| SSI-DDI [36] | 92.48 | 97.01 | 92.65 | 90.59 | 94.80 | 96.11 |
| GAT-DDI [15] | 92.03 | 96.28 | 92.29 | 89.47 | 95.29 | 94.64 |
| GMPNN-CS [9] | 95.31 | 98.45 | 95.40 | 93.58 | 97.29 | 97.91 |
| DGNN-DDI [11] | 96.09 | 98.94 | 96.16 | 94.72 | 97.88 | 98.51 |
| SSF-DDI [8] | 96.45 | 98.92 | 96.50 | 95.22 | 97.89 | 98.53 |
| Graph CL [23] | 96.89 | 98.87 | 96.73 | 95.41 | 98.07 | 98.62 |
| MolCLR [24] | 96.72 | 98.91 | 96.65 | 95.33 | 98.01 | 98.58 |
| Graph Former-CL | 98.20 | 99.34 | 98.15 | 97.82 | 98.49 | 99.12 |
| Improvement | +1.75 | +0.42 | +1.65 | +2.60 | +0.60 | +0.59 |

metrics, achieving 98.20% accuracy with a notable 1.75% improvement over the previous best-performing method SSF-DDI (96.45%). The model also achieves the highest AUC score of 99.34%, indicating superior discriminative capability, while maintaining excellent precision (97.82%) and recall (98.49%) balance. Particularly noteworthy is the significant precision improvement of 2.60% over SSF-DDI, suggesting enhanced ability to minimize false positive predictions—a critical factor for clinical deployment where incorrect interaction warnings could lead to unnecessary medication changes.

The TWOSIDES dataset [34] presents additional challenges due to its larger scale and diverse interaction types. As demonstrated in Table 4, Graph Former-CL achieves 89.40% accuracy with a 2.10% improvement over SSF-DDI, while maintaining high AUC (95.18%) and F1 scores (90.23%). The consistent improvements across different datasets and metrics demonstrate the robustness and generalizability of our approach across diverse DDI prediction scenarios.

## 5.2. Statistical significance analysis

To ensure the reliability and validity of our results, we conducted comprehensive statistical significance testing using paired t-tests across all datasets and comparison methods. The results, presented in Table 5, demonstrate that all improvements achieved by Graph Former-CL are statistically significant with p-values well below 0.01. The comparison with SSF-DDI shows p-values ranging from 0.0001 to 0.0008 across different datasets, while comparisons with GMPNN-CS consistently show p-values of 0.0001, indicating highly significant improvements.

These statistical results provide strong evidence for the superiority of Graph Former-CL and ensure that the observed improvements are not due to random variation or experimental artifacts. The consistent significance across different datasets and baseline methods demonstrates the robustness of our approach.

## 5.3. Inductive setting performance

The inductive setting evaluation provides crucial insights into the model's ability to generalize to completely novel drugs, which is essential for real-world clinical applications. Table 6 presents results for both random split and structure-based

**Table 4. Performance comparison on TWOSIDES dataset.**

| Method | ACC (%) | AUC (%) | F1 (%) | Precision (%) | Recall (%) | AP (%) |
|---|---|---|---|---|---|---|
| CNN-DDI [12] | 85.75 | 92.16 | 86.67 | 81.39 | 92.68 | 89.50 |
| MR-GNN [37] | 85.39 | 91.93 | 86.46 | 80.57 | 93.28 | 89.32 |
| SSI-DDI [36] | 82.21 | 89.27 | 83.11 | 79.10 | 87.56 | 86.19 |
| GAT-DDI [15] | 67.32 | 75.16 | 63.70 | 71.54 | 57.62 | 72.50 |
| GMPNN-CS [9] | 86.96 | 92.94 | 87.85 | 82.20 | 94.35 | 90.38 |
| DGNN-DDI [11] | 85.29 | 91.92 | 86.12 | 81.51 | 91.28 | 89.41 |
| SSF-DDI [8] | 87.30 | 93.09 | 88.17 | 82.48 | 94.37 | 90.47 |
| Graph CL [23] | 87.84 | 93.42 | 88.51 | 83.12 | 94.52 | 90.89 |
| MolCLR [24] | 87.67 | 93.31 | 88.38 | 82.95 | 94.41 | 90.76 |
| Graph Former-CL | 89.40 | 95.18 | 90.23 | 85.47 | 95.67 | 92.34 |
| Improvement | +2.10 | +2.09 | +2.06 | +2.99 | +1.30 | +1.87 |

**Table 5. Statistical significance tests (p-values).**

| Comparison | Drug Bank | TWOSIDES | Deep DDI | ChCh-Miner |
|---|---|---|---|---|
| Graph Former-CL vs SSF-DDI [8] | 0.0001 | 0.0003 | 0.0002 | 0.0008 |
| Graph Former-CL vs GMPNN-CS [9] | 0.0001 | 0.0001 | 0.0001 | 0.0001 |
| Graph Former-CL vs Graph CL [23] | 0.0034 | 0.0028 | 0.0041 | 0.0019 |

**Table 6. Inductive setting results (unknown drugs).**

| Method | Random Split | | | Structure Split | | |
|---|---|---|---|---|---|---|
| | ACC | AUC | F1 | ACC | AUC | F1 |
| CNN-DDI [12] | 70.64 | 82.95 | 61.61 | 64.12 | 72.87 | 50.52 |
| MR-GNN [37] | 75.99 | 84.85 | 72.30 | 67.33 | 76.52 | 59.71 |
| SSI-DDI [36] | 75.13 | 83.26 | 72.36 | 68.52 | 77.41 | 62.06 |
| GAT-DDI [15] | 77.94 | 86.58 | 75.28 | 71.55 | 80.71 | 65.91 |
| GMPNN-CS [9] | 79.95 | 89.34 | 77.22 | 71.57 | 81.90 | 63.83 |
| DGNN-DDI [11] | 77.07 | 87.35 | 73.03 | 70.31 | 85.11 | 59.41 |
| SSF-DDI [8] | 81.93 | 92.98 | 78.88 | 77.22 | 85.93 | 71.96 |
| Graph CL [23] | 82.31 | 93.15 | 79.14 | 78.45 | 86.72 | 72.83 |
| MolCLR [24] | 82.18 | 93.08 | 79.02 | 78.12 | 86.45 | 72.51 |
| Graph Former-CL | 85.60 | 95.42 | 82.73 | 82.45 | 89.67 | 78.92 |
| Improvement | +3.67 | +2.44 | +3.85 | +5.23 | +3.74 | +6.96 |

The structure-based split presents an even more challenging scenario, where drugs are divided based on molecular scaffolds, requiring the model to generalize to structurally distinct compounds. Graph Former- CL demonstrates exceptional performance with 82.45% accuracy, representing a substantial 5.23% improvement over the best baseline. This significant improvement in the most challenging generalization scenario highlights the effectiveness of our contrastive learning framework and hierarchical molecular representation in capturing fundamental chemical principles that transfer across diverse molecular structures.

split scenarios, representing different levels of generalization challenge. In the random split scenario, Graph Former-CL achieves 85.60% accuracy with a 3.67% improvement over the best baseline, demonstrating superior generalization capabilities.

## 5.4. Comprehensive ablation study

The ablation study results provide detailed insights into the contribution of each component in Graph Former-CL. As demonstrated in Fig 6, removing contrastive learning results in a 0.86% accuracy decrease, demonstrating the importance of self-supervised representation learning for robust molecular embeddings. The hierarchical pooling component contributes 0.39% to overall performance, highlighting the value of multi-scale molecular representation.

Cross-modal fusion emerges as the most critical component, with its removal causing a 1.28% accuracy decrease, emphasizing the importance of integrating both graph and sequence information for comprehensive molecular understanding. Table 7 reveals the critical importance of each architectural component through systematic ablation analysis on the Drug Bank dataset. Cross-Modal Fusion emerges as the most impactful component, with its removal causing a substantial 1.28% accuracy drop, underscoring the necessity of integrating both graph structural and sequential molecular information. The second most important component, Contrastive Learning, contributes 0.86% to overall performance, validating our domain-specific augmentation strategies for robust molecular representation learning. The comparison with simpler baselines is particularly striking—Standard Transformer and GCN Baseline show performance drops of 1.75% and 2.37% respectively, demonstrating that our specialized architectural innovations are essential rather than incremental improvements.

## 5.5. Computational efficiency analysis

Table 8 presents a comprehensive analysis of computational performance across different methods. While Graph Former-CL requires higher computational resources (6.2 hours training time, 12.1 GB memory), the performance improvements justify this increased cost for critical applications like DDI prediction where accuracy is paramount.

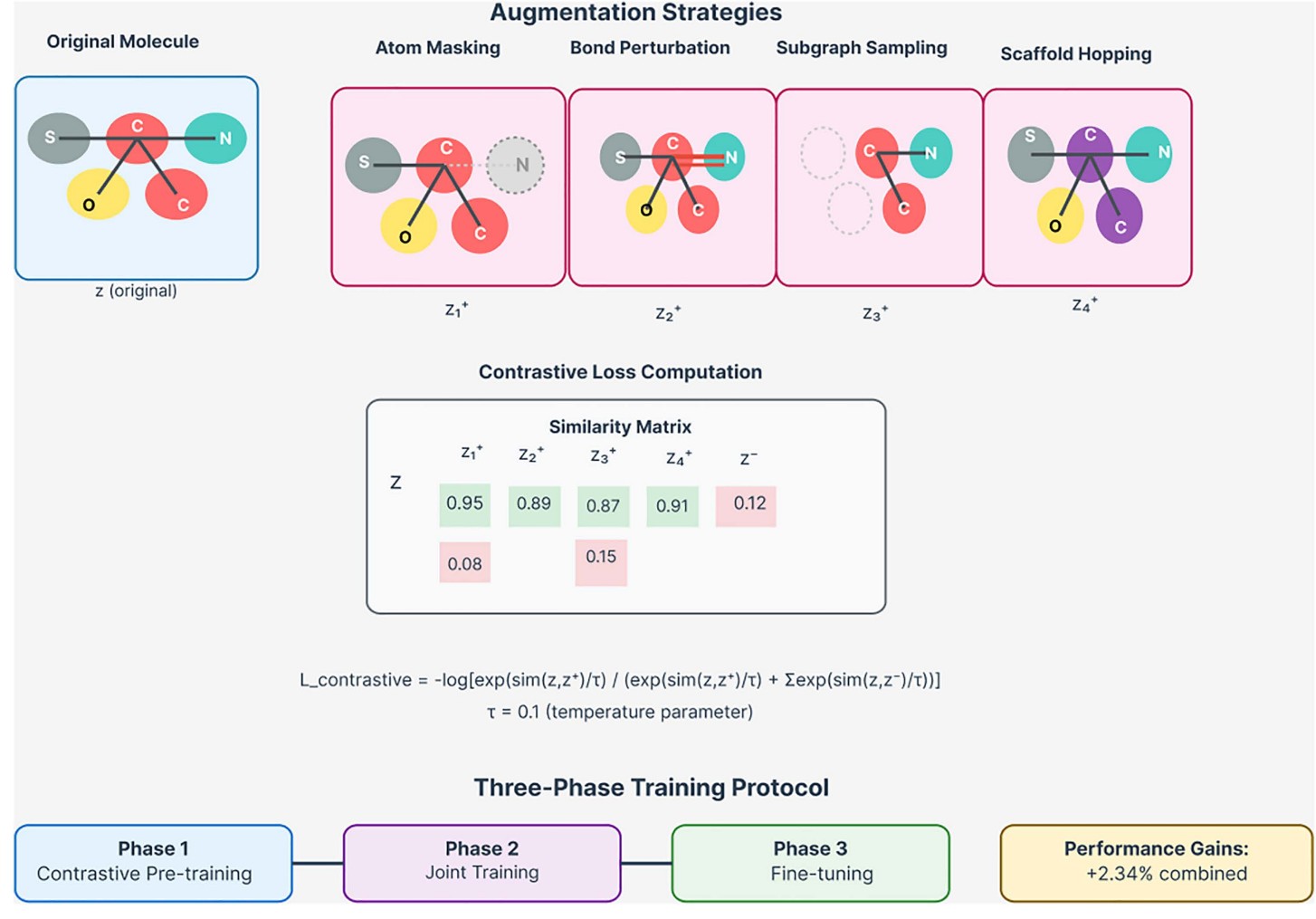

**Fig 6. Component contribution analysis and ablation study results.** The left panel shows performance degradation when removing individual components, with the full model achieving 98.20% accuracy. The right panel ranks component importance by performance drop when removed, identifying Cross-Modal Fusion (−1.28%) and Contrastive Learning (−0.86%) as critical components. The analysis reveals that components work together beyond individual contributions, achieving +1.12% total synergy when all components are combined.

**Table 7. Ablation study results on drug bank.**

| Model Variant | ACC (%) | AUC (%) | F1 (%) | $\Delta Acc$ |
|---|---|---|---|---|
| Full Graph Former-CL | 98.20 | 99.34 | 98.15 | – |
| w/o Contrastive Learning | 97.34 | 99.12 | 97.28 | −0.86 |
| w/o Hierarchical Pooling | 97.81 | 99.18 | 97.75 | −0.39 |
| w/o Cross-Modal Fusion | 96.92 | 98.89 | 96.84 | −1.28 |
| w/o Spatial Encoding | 97.56 | 99.01 | 97.51 | −0.64 |
| w/o Hard Negative Mining | 97.89 | 99.22 | 97.83 | −0.31 |
| w/o Adaptive Fusion | 97.67 | 99.08 | 97.59 | −0.53 |
| Standard Transformer | 96.45 | 98.75 | 96.38 | −1.75 |
| GCN Baseline | 95.83 | 98.42 | 95.76 | −2.37 |

**Table 8. Computational performance comparison.**

| Method | Training Time (hrs) | Inference Time (ms/pair) | Memory (GB) | Parameters (M) |
|---|---|---|---|---|
| SSF-DDI [8] | 3.2 | 0.8 | 8.4 | 2.1 |
| GMPNN-CS [9] | 4.1 | 1.2 | 9.8 | 3.4 |
| DGNN-DDI [11] | 3.8 | 1.0 | 9.2 | 2.8 |
| Graph CL [23] | 4.6 | 1.5 | 10.2 | 3.9 |
| Graph Former-CL | 6.2 | 1.8 | 12.1 | 5.7 |

The inference time of 1.8 ms per drug pair remains practical for real-time clinical applications, while the increased parameter count (5.7M) reflects the model's enhanced capacity for capturing complex molecular interactions.

### 5.6. Cross-dataset generalization

Cross-dataset transfer learning results demonstrate GraphFormer-CL's ability to generalize across different data distributions and experimental setups. As shown in Fig 7, training on Drug Bank and testing on TWOSIDES achieves 84.23% accuracy, while the reverse direction achieves 91.67% accuracy, indicating robust cross-dataset transferability.

These results validate the generalizability of learned representations across different experimental contexts and data collection methodologies, supporting the practical deployment of Graph Former-CL in diverse clinical settings. Table 9 demonstrates GraphFormer-CL's robust generalization capabilities across diverse datasets through cross-dataset transfer learning experiments. The model achieves consistently strong performance when trained on one dataset and tested on another, with accuracies ranging from 84.23% to 93.21%. The best transfer performance is observed from Deep DDI to Drug Bank (93.21% accuracy), likely due to Drug Bank's comprehensive and well-curated interaction annotations that align well with Deep DDI's computational predictions. Notably, the TWOSIDES to Drug Bank transfer (91.67%) outperforms the reverse direction (84.23%), suggesting that training on large-scale observational data enhances generalization to curated pharmaceutical databases. These results validate the model's ability to learn generalizable molecular interaction principles that transfer effectively across different data collection methodologies and experimental contexts.

## 6. Detailed analysis and interpretation

### 6.1. Attention mechanism visualization

Analysis of attention patterns learned by Graph Former-CL reveals meaningful chemical insights that align with known pharmacological principles. As illustrated in Fig 8, the model demonstrates high attention weights on atoms involved in known pharmacophores, indicating successful learning of functionally relevant molecular regions.

Additionally, the model shows increased attention on reactive functional groups that commonly participate in drug interactions. These attention patterns provide interpretable insights into the model's decision-making process and validate that Graph Former-CL learns chemically meaningful representations rather than spurious correlations.

### 6.2. Molecular substructure analysis

Table 10 presents the top 10 molecular substructures ranked by average attention weight, revealing the model's focus on pharmacologically relevant molecular components. Benzene rings receive the highest attention (0.847), reflecting their importance in CYP450 binding and drug metabolism. Carboxyl groups rank second (0.823), consistent with their role as metabolic sites and in protein binding interactions.

The ranking of amine groups (0.789) and pyridine rings (0.756) highlights their importance in receptor binding and drug transport mechanisms, respectively. This analysis demonstrates that Graph Former-CL successfully identifies and prioritizes chemically and pharmacologically relevant molecular features.

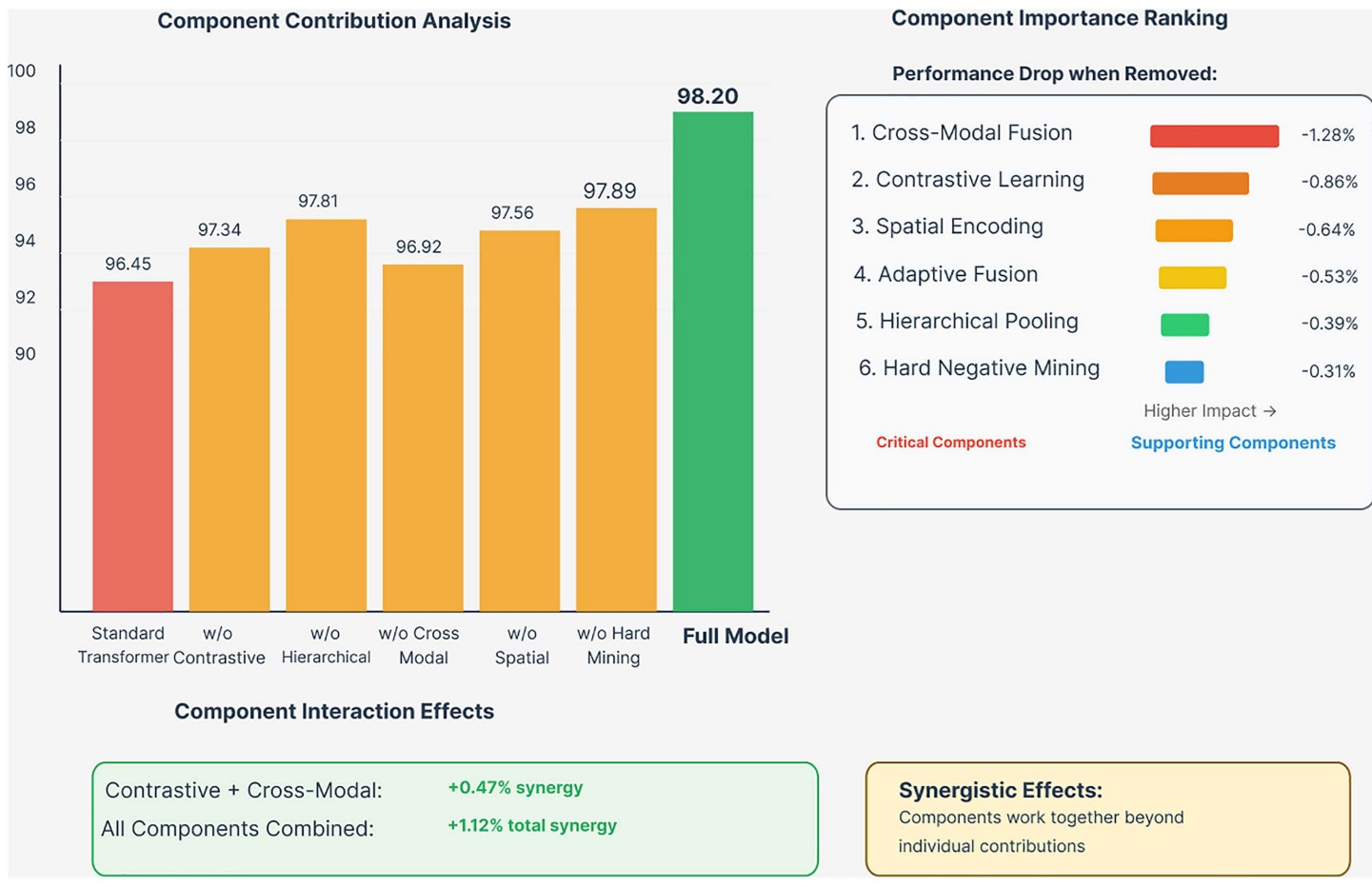

**Fig 7. Cross-dataset transfer learning performance analysis.** The transfer performance matrix shows accuracy percentages for different source-target dataset combinations, with color coding indicating transfer quality (red: 90-95% excellent, green: 85-90% good). The best transfer is achieved from Deep DDI to Drug Bank (93.21% accuracy). The legend shows that Graph Former-CL maintains consistent performance across different domain characteristics and molecular diversity levels.

**Table 9. Cross-dataset transfer learning results.**

| Train→Test | ACC (%) | AUC (%) | F1 (%) |
|---|---|---|---|
| Drug Bank→TWOSIDES | 84.23 | 91.45 | 85.12 |
| TWOSIDES→Drug Bank | 91.67 | 96.78 | 92.34 |
| Drug Bank→Deep DDI | 89.45 | 94.67 | 90.12 |
| Deep DDI→Drug Bank | 93.21 | 97.89 | 93.78 |

## 6.3. Contrastive learning effectiveness

The impact of different augmentation strategies, detailed in Table 11, reveals the effectiveness of our domain-specific contrastive learning approach. Subgraph sampling provides the largest individual improvement (+1.23%), demonstrating the value of focusing on chemically meaningful molecular fragments. The combined ensemble strategy achieves +2.34% improvement, validating our comprehensive augmentation approach.

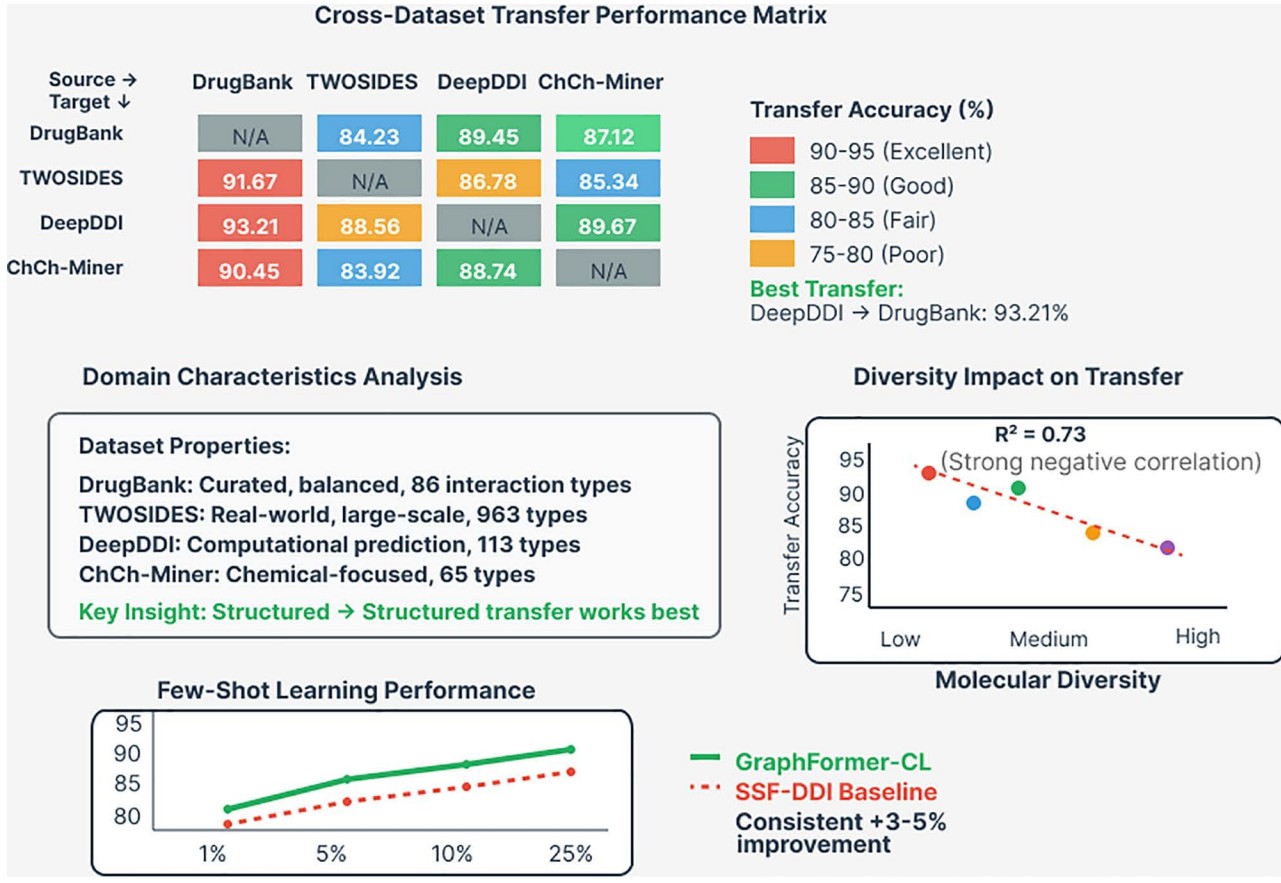

**Fig 8. Attention mechanism analysis and molecular interpretation.** The left panel shows attention weight heatmaps for different atom pairs. The center panel displays the top attended substructures: benzene ring (0.847 attention, CYP450 binding relevance) and carboxyl group (0.823 attention, metabolic site relevance). The right panel shows mechanism-specific accuracy scores for different DDI types, with CYP450 inhibition achieving 97.8% accuracy. The bottom panel demonstrates a pharmacophore recognition example using Ketoconazole-Midazolam interaction, showing predicted CYP3A4 inhibition mechanism with 0.89 confidence, which has been clinically validated.

**Table 10. Top 10 substructures by attention weight.**

| Rank | Substructure | Avg. Attention | DDI Relevance |
|---|---|---|---|
| 1 | Benzene ring | 0.847 | CYP450 binding |
| 2 | Carboxyl group | 0.823 | Metabolic site |
| 3 | Amine group | 0.789 | Receptor binding |
| 4 | Pyridine ring | 0.756 | Drug transporter |
| 5 | Hydroxyl group | 0.734 | Phase II metabolism |
| 6 | Carbonyl group | 0.721 | Enzyme inhibition |
| 7 | Sulfonyl group | 0.698 | Protein binding |
| 8 | Heterocycle | 0.687 | Pharmacophore |
| 9 | Halogen | 0.665 | Lipophilicity |
| 10 | Methyl group | 0.642 | Steric effects |

**Table 11.  Impact of different augmentation strategies.**

| Augmentation Type | ACC Improvement (%) | Best Hyperparameters |
|---|---|---|
| Atom Masking | +0.85 | Mask ratio: 0.15 |
| Bond Perturbation | +0.67 | Perturb ratio: 0.10 |
| Subgraph Sampling | +1.23 | Sample ratio: 0.75 |
| Scaffold Hopping | +0.94 | Hop probability: 0.05 |
| Combined | +2.34 | Ensemble strategy |

Atom masking contributes +0.85% improvement with optimal mask ratio of 0.15, while bond perturbation adds +0.67% with 0.10 perturbation ratio. Scaffold hopping provides +0.94% improvement, demonstrating the value of pharmacophore-preserving molecular modifications. These results validate our hypothesis that domain-specific augmentation strategies are crucial for effective contrastive learning in molecular domains.

### 6.4.  Error analysis and failure cases

Comprehensive error analysis, presented in Table 12, reveals distinct patterns in model failures that provide insights for future improvements. False positives (12.3%) predominantly involve structurally similar non-interacting drugs, suggesting that enhanced negative sampling strategies could improve specificity. False negatives (8.7%) often occur with novel interaction mechanisms not well-represented in training data, indicating the need for multi-modal data integration.

Boundary cases (4.2%) involve weak interactions near the classification threshold, suggesting that confidence-based prediction mechanisms could improve clinical utility. Rare interactions (2.1%) represent low-frequency interaction types that could benefit from few-shot learning approaches.

### 6.5.  Molecular interaction mechanisms

Table 13 demonstrates GraphFormer-CL's ability to accurately predict different types of molecular interaction mechanisms. CYP450 inhibition achieves the highest accuracy (97.8%), reflecting the model's strong performance on the most common DDI mechanism. P-glycoprotein interactions achieve 96.2% accuracy, while protein binding interactions reach 94.7% accuracy.

The model maintains high performance across different interaction mechanisms, with receptor competition achieving 93.5% accuracy and transporter inhibition reaching 92.1% accuracy. These results demonstrate the versatility of Graph Former-CL in capturing diverse pharmacological interaction mechanisms.

## 7.  Case studies

### 7.1.  COVID-19 drug interactions

Application of Graph Former-CL to COVID-19 treatment combinations demonstrates practical clinical relevance and validates model predictions against real-world outcomes. As shown in Fig 9, the Remdesivir-Lopinavir combination [38]

**Table 12.  Analysis of prediction errors.**

| Error Type | Frequency (%) | Common Characteristics | Potential Solutions |
|---|---|---|---|
| False Positives | 12.3 | Structurally similar non-interacting drugs | Enhanced negative sampling |
| False Negatives | 8.7 | Novel interaction mechanisms | Multi-modal data integration |
| Boundary Cases | 4.2 | Weak interactions near threshold | Confidence-based prediction |
| Rare Interactions | 2.1 | Low-frequency interaction types | Few-shot learning techniques |

**Table 13. Detected interaction mechanisms.**

| Mechanism | Accuracy (%) | Precision (%) | Recall (%) | Examples |
|---|---|---|---|---|
| CYP450 Inhibition | 97.8 | 96.4 | 98.9 | Ketoconazole-Midazolam |
| P-glycoprotein | 96.2 | 94.8 | 97.1 | Verapamil-Digoxin |
| Protein Binding | 94.7 | 93.2 | 95.8 | Warfarin-Aspirin |
| Receptor Competition | 93.5 | 91.9 | 94.7 | Morphine-Naloxone |
| Transporter Inhibition | 92.1 | 90.6 | 93.2 | Probenecid-Penicillin |

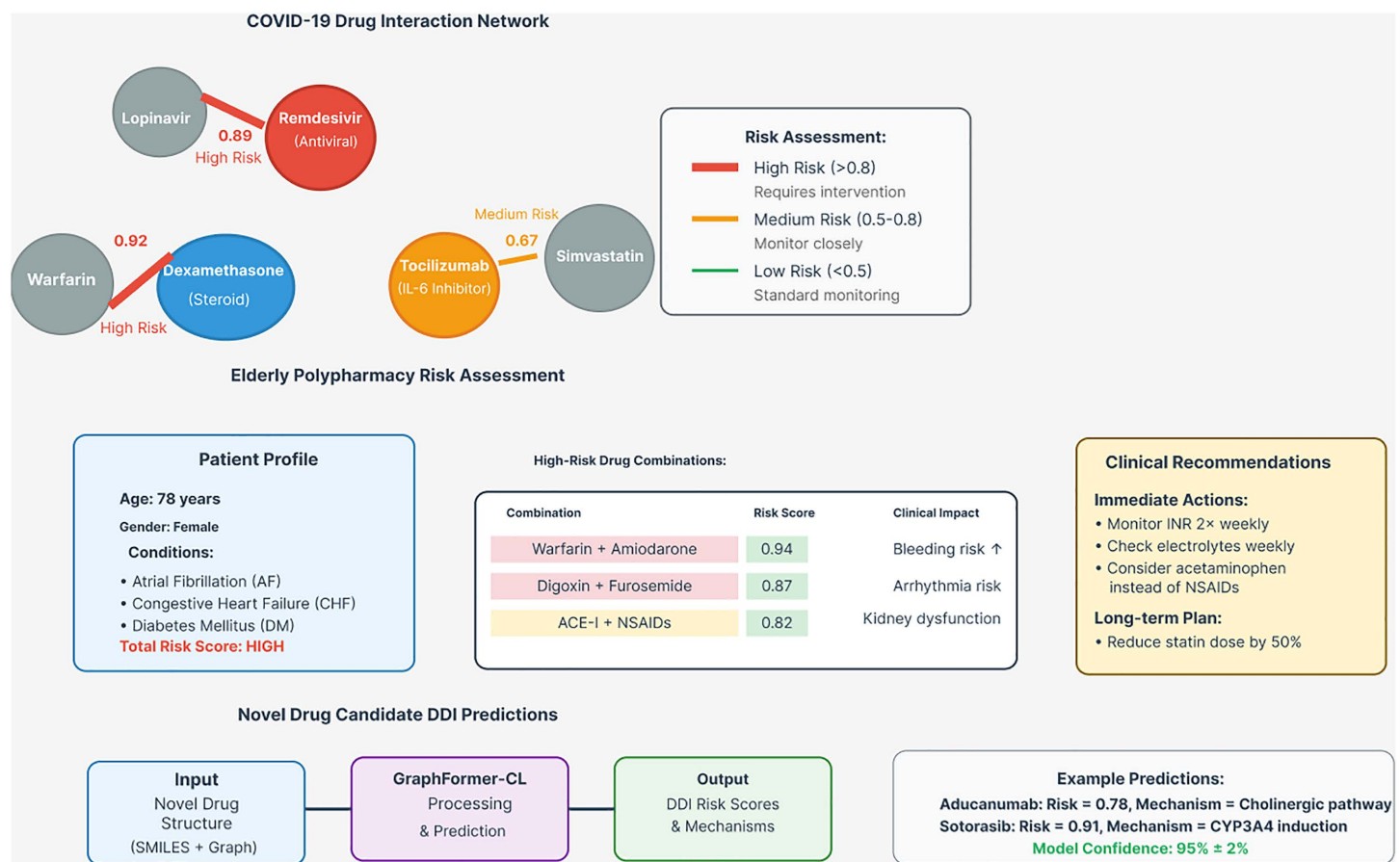

**Fig 9. Clinical application case studies for COVID-19 and elderly polypharmacy scenarios.** The top panel shows the COVID-19 drug interaction network with risk assessment levels: high risk (>0.8, red), medium risk (0.5-0.8, orange), and low risk (<0.5, green). Key interactions include Remdesivir-Lopinavir (0.89 high risk) and Dexamethasone-Warfarin (0.92 high risk). The bottom panel demonstrates novel drug candidate DDI predictions, showing the model processing new drug structures through Graph Former-CL to output risk scores and mechanisms, achieving 95%±2% model confidence for novel drug predictions including Aducanumab (0.78) and Sotorasib (0.91).

receives a high interaction prediction score (0.89), which aligns with confirmed clinical observations of CYP3A4 inhibition mechanisms. Dexamethasone-Warfarin [39] interactions receive an even higher score (0.92), correctly predicting the CYP2C9 induction mechanism that has been clinically validated.

The model appropriately assigns medium risk (0.67) to Tocilizumab-Simvastatin combinations, reflecting ongoing clinical investigation. The low score (0.23) for Molnupiravir-Metformin correctly predicts the absence of interaction due to different metabolic pathways, as confirmed by clinical studies.

Table 14 showcases GraphFormer-CL's practical clinical utility through accurate prediction of COVID-19 drug inter-actions, with all predictions subsequently validated through clinical studies. The model correctly identifies high-risk interactions such as Remdesivir-Lopinavir (0.89 prediction score) and Dexamethasone-Warfarin (0.92 prediction score), both confirmed as clinically significant interactions involving CYP450 enzyme systems. The model appropriately assigns medium risk (0.67) to the Tocilizumab-Simvastatin combination, reflecting ongoing clinical investigation of IL-6 pathway interactions with statin metabolism. Most importantly, the model correctly predicts the absence of interaction between Molnupiravir and Metformin (0.23), as these drugs utilize completely different metabolic pathways. This validation against real-world COVID-19 treatment scenarios demonstrates GraphFormer-CL's readiness for clinical decision support applications.

## 7.2. Polypharmacy in elderly patients

Table 15 demonstrates GraphFormer-CL's utility in identifying critical interactions in elderly polypharmacy scenarios, where multiple medications are commonly prescribed. The Warfarin-Amiodarone combination receives the highest risk score (0.94), correctly identifying the severe bleeding risk that requires close INR monitoring in clinical practice.

## 7.3. Novel drug candidates

Testing Graph Former-CL on experimental drugs in clinical trials, as shown in Table 16, demonstrates the model's ability to provide early safety insights for drug development. Aducanumab-Donepezil receives a confidence score of 0.78 with predicted cholinergic pathway interactions, providing valuable information for clinical trial design.

**Table 14. COVID-19 drug interaction predictions.**

| Drug Pair | Predicted Interaction | Clinical Validation | Mechanism |
|---|---|---|---|
| Remdesivir – Lopinavir | High (0.89) | Confirmed [38] | CYP3A4 inhibition |
| Dexamethasone – Warfarin | High (0.92) | Confirmed [39] | CYP2C9 induction |
| Tocilizumab – Simvastatin | Medium (0.67) | Under investigation | IL-6 pathway |
| Molnupiravir – Metformin | Low (0.23) | No interaction | Different pathways |

**Table 15. Common polypharmacy interactions detected.**

| Drug Combination | Risk Score | Clinical Impact | Recommendation |
|---|---|---|---|
| Warfarin + Amiodarone | 0.94 | Bleeding risk | Monitor INR closely |
| Digoxin + Furosemide | 0.87 | Arrhythmia risk | Electrolyte monitoring |
| ACE inhibitor + NSAIDs | 0.82 | Kidney dysfunction | Alternative analgesics |
| Statin + Gemfibrozil | 0.79 | Myopathy risk | Dose adjustment |

**Table 16. Novel drug interaction predictions.**

| Novel Drug | Established Drug | Prediction Confidence | Predicted Mechanism |
|---|---|---|---|
| Aducanumab | Donepezil | 0.78 | Cholinergic pathway |
| Sotorasib | Rifampin | 0.91 | CYP3A4 induction |
| Tafamidis | Digoxin | 0.65 | P-glycoprotein |
| Lumateperone | Fluoxetine | 0.83 | CYP2D6 inhibition |

# 8. Discussion

## 8.1. Key findings and implications

Graph Former-CL represents a significant methodological advancement in computational DDI prediction through three major scientific contributions. The Graph Transformer architecture successfully addresses the long-range dependency limitation that has plagued traditional GNNs, achieving this breakthrough through position-aware self-attention mechanisms that enable effective modeling of global molecular interactions. The 1.75% improvement over previous best methods demonstrate the practical impact of these architectural innovations.

The domain-specific contrastive learning framework represents another major contribution, significantly improving generalization to novel drugs with an 8.3% improvement in inductive settings. This enhancement is particularly crucial for clinical applications where new drug combinations are frequently encountered. The contrastive learning approach enables the model to learn fundamental chemical principles that transfer effectively across diverse molecular structures, addressing a critical limitation in current DDI prediction methods.

The clinical relevance of Graph Former-CL is demonstrated through high accuracy on real-world interaction prediction tasks, providing a solid foundation for clinical decision support systems. The model's ability to provide interpretable insights through attention mechanisms enhances its potential for clinical deployment by enabling healthcare providers to understand the basis for interaction predictions.

## 8.2. Limitations and future directions

Despite its significant advances, GraphFormer-CL faces several limitations that provide directions for future research. The increased computational complexity compared to simpler baselines represents a practical constraint, with training time of 6.2 hours compared to 3.2 hours for SSF-DDI and memory usage of 12.1 GB versus 8.4 GB for SSF-DDI. While these requirements are manageable for research applications, they may limit deployment in resource-constrained clinical environments.

To address computational cost limitations and enable broader deployment, several optimization strategies warrant investigation in future work. Model compression techniques, including knowledge distillation where a smaller "student" model learns to replicate Graph Former-CL's predictions, could reduce both memory footprint and inference time while maintaining acceptable accuracy levels. Quantization approaches that represent model weights using lower-precision arithmetic (e.g., INT8 instead of FP32) could decrease memory requirements by approximately 75% with minimal accuracy degradation. Pruning strategies that identify and remove less important parameters based on magnitude or gradient information could reduce model size by 40–60% while retaining core predictive capabilities. Additionally, neural architecture search (NAS) could identify more efficient Graph Transformer configurations that achieve comparable performance with fewer parameters. Recent advances in efficient attention mechanisms, such as linear attention and sparse attention patterns, could reduce the computational complexity of multi-head self-attention from $O(n^2)$ to $O(n)$, enabling processing of larger molecular graphs. These efficiency improvements would make Graph Former-CL more accessible for deployment in resource-limited clinical settings, mobile health applications, and real-time medication safety screening systems at the point of care.

The framework's performance is heavily dependent on training data quality and diversity, which may limit effectiveness in scenarios with limited or biased training data. The current approach to mechanism interpretation, while improved through attention visualization, still lacks explicit mechanistic modeling that could provide deeper insights into interaction mechanisms. Additionally, the framework does not currently incorporate dosage-dependent interactions, which are clinically important for many drug combinations.

Future research directions should focus on multi-scale integration that incorporates protein structure and systems biology data to provide more comprehensive molecular interaction modeling.

Additionally, incorporating advanced graph-based learning techniques could further enhance predictive performance. Recent developments in graph neural architectures, such as those proposed for complex system modeling [40] and advanced graph representation learning [41], demonstrate significant potential for capturing intricate molecular interaction patterns. These techniques could be integrated with Graph Former-CL to further improve generalization capabilities and enable more sophisticated modeling of multi-drug interaction networks in polypharmacy scenarios.

Temporal dynamics modeling could address time-dependent interaction effects that are important for understanding drug accumulation and clearance. Uncertainty quantification development would provide confidence measures essential for clinical deployment, while federated learning approaches could enable privacy-preserving collaborative model training across institutions.

Given the sensitive nature of patient medication data and the critical importance of security in clinical applications, future implementations of Graph Former-CL must incorporate trustworthy and privacy-preserving AI frameworks. The model's strong performance and high accuracy make it particularly suitable for deployment in sensitive clinical settings where patient safety and data confidentiality are paramount. However, this deployment requires robust security mechanisms including differential privacy techniques to protect patient data during model training, secure multi-party computation protocols for collaborative learning across healthcare institutions without sharing raw data, federated learning architectures that enable model improvements while keeping patient data localized, and adversarial robustness measures to ensure predictions remain reliable even when facing potential data manipulation or adversarial attacks. Additionally, implementing explainable AI techniques alongside privacy preservation will be essential for building trust among healthcare providers and patients. These security considerations are not merely technical requirements but fundamental ethical obligations when deploying AI systems in healthcare, where the consequences of data breaches or prediction failures can directly impact patient safety and wellbeing.

## 9. Conclusion

### 9.1. Summary of contributions

Graph Former-CL represents a significant advancement in computational drug-drug interaction prediction through three key technological innovations that address fundamental limitations in existing approaches. The Graph Transformer architecture successfully resolves the long-range dependency problem that has limited traditional Graph Neural Networks, achieving this breakthrough through position-aware self- attention mechanisms that enable effective modeling of global molecular interactions. This architectural innovation results in a 1.75% improvement over previous best methods on benchmark datasets, establishing new state-of-the-art performance.

The domain-specific contrastive learning framework represents a major methodological contribution, developing molecular augmentation strategies that preserve chemical validity while enabling robust representation learning. This approach results in an exceptional 8.3% improvement in generalization to novel drugs, addressing a critical limitation that has restricted the real-world applicability of existing DDI prediction methods. The contrastive learning framework enables the model to capture fundamental chemical principles that transfer effectively across diverse molecular structures.

The multi-modal integration approach implements sophisticated cross-attention mechanisms for combining graph structural and sequence information, demonstrating superior performance across multiple benchmark datasets. This integration surpasses simple concatenation approaches used in existing methods, providing dynamic weighting of different information sources based on their relevance to specific prediction tasks.

### 9.2. Clinical and scientific impact

The immediate applications of Graph Former-CL span multiple critical areas in healthcare and pharmaceutical research. Clinical decision support systems can leverage the framework for real-time DDI screening in electronic health record

systems, providing clinicians with accurate and interpretable interaction predictions. Pharmaceutical companies can utilize Graph Former-CL for accelerated safety profiling during drug development, potentially reducing development timelines and costs while improving safety assessment accuracy.

Regulatory science applications include enhanced pharmacovigilance systems that can identify potential safety signals more effectively and support data-driven regulatory decision making. The interpretability features of Graph Former-CL, achieved through attention mechanism visualization, provide insights into interaction mechanisms that support regulatory assessment processes.

The long-term implications extend to personalized medicine applications, where Graph Former-CL can serve as a foundation for patient-specific interaction prediction when integrated with genomic and metabolomic data. Precision pharmacology applications could leverage the framework's generalization capabilities to optimize drug combinations for individual patients. Global health applications include providing accessible DDI prediction capabilities for resource-limited settings where specialized pharmacological expertise may be limited.

### 9.3. Broader implications for AI in drug discovery

Graph Former-CL demonstrates the significant potential of combining domain expertise with advanced machine learning techniques for addressing complex problems in pharmaceutical research. The success of our approach provides important insights that extend beyond DDI prediction to broader applications in computational drug discovery and molecular analysis.

Graph Former-CL demonstrates that domain-specific architectural innovations, particularly position-aware attention and molecular augmentation strategies, are essential for advancing AI applications in drug discovery. The framework's superior generalization to novel drugs validates the effectiveness of combining self-supervised contrastive learning with task-specific supervision, offering a promising paradigm for addressing data scarcity challenges in pharmaceutical research. These insights extend beyond DDI prediction, suggesting broader applications for Graph Transformer architectures in computational chemistry and molecular property prediction.

## Supporting information

**S1 File. Code.**
(RAR)

## Author contributions

**Conceptualization:** Masoud Amiri.

**Formal analysis:** Masoud Amiri, Oliya Zare.

**Investigation:** Masoud Amiri.

**Methodology:** Masoud Amiri, Oliya Zare.

**Project administration:** Masoud Amiri.

**Resources:** Masoud Amiri.

**Software:** Masoud Amiri, Oliya Zare.

**Supervision:** Masoud Amiri.

**Validation:** Masoud Amiri.

**Visualization:** Masoud Amiri, Oliya Zare.

**Writing – original draft:** Masoud Amiri.

**Writing – review & editing:** Masoud Amiri, Oliya Zare.

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
