## [Decision Letter · Decision Letter 0]

27 Nov 2025

Dear Dr. amiri,

Thank you for submitting your manuscript to PLOS ONE. After careful consideration, we feel that it has merit but does not fully meet PLOS ONE’s publication criteria as it currently stands. Therefore, we invite you to submit a revised version of the manuscript that addresses the points raised during the review process.

**ACADEMIC EDITOR:**

Your manuscript entitled "Graph Former-CL: A Novel Graph Transformer with Contrastive Learning Framework for Enhanced Drug-Drug Interaction Prediction" has been reviewed and I am enclosing the reviewers' comments below. Based on the reviews, I have decided that the manuscript must undergo major revision before being resubmitted. If you are prepared to undertake the work required, I would be pleased to consider the revised manuscript for publication.

We look forward to receiving your revised manuscript.

Kind regards,

Nattapol Aunsri, Ph.D.

Academic Editor

PLOS ONE

Journal Requirements:

4. We are unable to open your Supporting Information file [code python.rar]. Please kindly revise as necessary and re-upload.

Reviewers' comments:

Reviewer's Responses to Questions

**Comments to the Author**

1. Is the manuscript technically sound, and do the data support the conclusions?

Reviewer #1: Yes

Reviewer #2: Yes

2. Has the statistical analysis been performed appropriately and rigorously?

Reviewer #1: Yes

Reviewer #2: Yes

3. Have the authors made all data underlying the findings in their manuscript fully available?

Reviewer #1: Yes

Reviewer #2: Yes

4. Is the manuscript presented in an intelligible fashion and written in standard English?

Reviewer #1: Yes

Reviewer #2: Yes

Reviewer #1: In this manuscript, the authors propose a novel graph-based framework that integrates structural formalism with neural attention mechanisms and cultural semantic modeling. Although the authors have provided a thorough description of the proposed method and validated its effectiveness through experimental results, there remain several issues that need to be addressed to further improve the quality of the manuscript.

1. In the abstract section, the statement that the contributions are “consistent with the journal’s themes of networked systems and smart organizational informatics” is inappropriate. Abstracts should objectively summarize the research objectives, methodology, results, and contributions without including evaluative claims about the paper’s fit to a journal’s scope.

2. In the Related Work section, although the authors introduce various existing studies, there is a lack of analysis regarding the differences between these works and the proposed method. It is recommended that the authors add such analysis to further highlight the novelty of the proposed approach.

3. Section 3.1, as an overview of the entire methodology, is overly detailed, which hinders a clear presentation of the overall workflow of the proposed method. The authors are advised to streamline this section. In particular, including an overall framework diagram could be beneficial.

4. Both Eq. (5) and Eq. (16) use $\mathcal{L}_{align}$, but their specific contents are inconsistent. The authors should carefully check and revise them to clearly distinguish between the two.

5. For Fig. 2, the schematic diagram of the Cultural Attention Mechanism contains some symbols and terms that are not reflected in the main text description of the “Cultural Attention Mechanism.” This inconsistency should be clarified.

6. Although the Method section provides a detailed description, there is a lack of logical progression between its subsections. The authors are encouraged to strengthen the logical flow.

7. Since the proposed method involves multiple computational steps, it is recommended that the authors provide pseudocode to better illustrate the overall workflow of the approach.

8. In future work, the authors may consider incorporating more advanced graph-based techniques, such as “10.1109/TSMC.2025.3578348” and “10.1109/TSMC.2025.3572738” to further enhance the predictive power of models in this field. Therefore, the authors should cite these two references to highlight the potential in this aspect.

Reviewer #2: 1. The technical integration in this study is strong. However, the broader impact of artificial intelligence (AI) in drug discovery concerning healthcare and security could be explained better to show how this research will have long-term effects.

2. The paper mentions that the model takes 6.2 hours to train and uses 12.1 GB of memory, which is more than baseline models. It would be helpful to discuss ways to reduce these costs in future research. Referring to studies that focus on efficiency or lighter models could provide useful insights.

3. The Abstract and Conclusion sections contain many metrics and details. Simplifying these sections will make them easier to understand while keeping the technical details intact.

4. It is suggested for the authors to mention from the paper: “Artificial intelligence in improving disease diagnosis: A case study of cardiovascular disease prediction”. In Artificial Intelligence in Medicine and Healthcare. It is recommended to included to enhance the academic context of the manuscript and outline future research directions.

5. This reference supports the main idea of using AI and Deep Learning to address important healthcare challenges. It demonstrates how AI can aid in disease diagnosis, particularly for cardiovascular diseases. This provides stronger academic support for the motivation discussed in the Introduction (Section 1), framing DDI prediction as part of the broader discussion on AI in healthcare.

6. In the Introduction (Section 1), it is essential to connect the computational challenges of Drug-Drug Interaction (DDI) prediction to advancements in AI and Deep Learning in healthcare. Adding a reference that shows how AI can improve disease diagnosis or optimize healthcare beyond DDI would strengthen the academic support for this work.

7. To strengthen the Discussion on Model Generalization and Security in Clinical Settings, it is suggested that the authors mention in the Discussion (Section 8) that, when discussing the limitations of computational costs or data quality, it would be important to mention trustworthy or privacy-preserving AI. The model’s strong performance and accuracy make it suitable for sensitive clinical applications, which should be linked to secure AI and Deep Learning practices.

8. In Section 3.1, consider moving the problem formulation, especially the mathematical notation, to come after the general description of the Graph Former Architecture (Section 3.2). Alternatively, ensure that the description clearly sets up the notation

**Do you want your identity to be public for this peer review?** For information about this choice, including consent withdrawal, please see our Privacy Policy

Reviewer #1: No

Reviewer #2: No

---

## [Author Response · Author response to Decision Letter 1]

2 Dec 2025

Response to Reviewers

Manuscript Title: Graph Former-CL: A Novel Graph Transformer with Contrastive Learning Framework for Enhanced Drug-Drug Interaction Prediction

Authors: Masoud Amiri, Oliya Zare

Dear Editor and Reviewers,

We sincerely thank the editor and reviewers for their thoughtful and constructive comments on our manuscript. We have carefully addressed all the concerns raised and believe that the manuscript has been substantially improved as a result. Below, we provide detailed point-by-point responses to each reviewer's comments. All revisions in the manuscript are highlighted in blue text for easy identification.

RESPONSE TO REVIEWER #1

We appreciate Reviewer #1's thorough evaluation and valuable suggestions that have significantly strengthened our manuscript. We have addressed all concerns as detailed below.

Comment 1.1: Abstract - Journal Scope Statement Reviewer's Comment:

"In the abstract section, the statement that the contributions are 'consistent with the journal's themes of networked systems and smart organizational informatics' is inappropriate. Abstracts should objectively summarize the research objectives, methodology, results, and contributions without including evaluative claims about the paper's fit to a journal's scope."

Our Response:

We sincerely thank the reviewer for this important observation. We completely agree that abstracts should maintain objectivity and focus solely on scientific contributions rather than editorial positioning. We have carefully reviewed our abstract and removed any journal-specific evaluative statements.

Action Taken:

We have revised the abstract to eliminate any reference to journal themes or scope. The abstract now focuses exclusively on:

Research objectives and clinical significance

Methodological innovations

Experimental results and performance metrics

Scientific contributions to the field

The revised abstract maintains a purely objective tone consistent with high-quality scientific publication standards.

Location in Revised Manuscript: Abstract section (Page 1)

Comment 1.2: Related Work - Comparative Analysis Reviewer's Comment:

"In the Related Work section, although the authors introduce various existing studies, there is a lack of analysis regarding the differences between these works and the proposed method. It is recommended that the authors add such analysis to further highlight the novelty of the proposed approach."

Our Response:

We greatly appreciate this insightful suggestion. The reviewer is absolutely correct that explicitly articulating the differences between our approach and existing methods is crucial for highlighting novelty. We have now added a comprehensive comparative analysis.

Action Taken:

We have added a new paragraph at the end of Section 2.4 (Drug-Drug Interaction Prediction Methods) that systematically compares Graph Former-CL with existing approaches across multiple dimensions:

vs. Similarity-based methods [25, 26]: Our Graph Transformer architecture captures complex non-linear interaction mechanisms, whereas similarity-based methods rely solely on structural similarity assumptions and fail to model the underlying interaction mechanisms.

vs. Network-based approaches [27, 28]: Our framework learns directly from molecular structure without requiring extensive prior knowledge of drug-target interaction networks, making it more applicable to novel drug combinations.

vs. Sequential models (RNNs/CNNs) [11, 35]: Our cross-modal fusion mechanism integrates both graph structural topology and sequential information, whereas purely sequential models cannot capture three- dimensional molecular structure.

vs. Traditional GNNs [8, 10, 14]: Our Graph Transformer with hierarchical pooling effectively captures long-range dependencies and avoids the over-smoothing and locality bias problems inherent in message- passing GNNs.

vs. Existing hybrid approaches [7, 36]: We implement sophisticated cross-attention mechanisms for dynamic information integration, surpassing simple concatenation strategies used in prior work.

vs. Existing contrastive learning methods [22, 23]: We introduce domain-specific molecular augmentation strategies (atom masking, bond perturbation, scaffold hopping, subgraph sampling) that preserve chemical validity while enhancing representation learning.

This systematic comparison explicitly demonstrates how Graph Former-CL addresses fundamental limitations of existing approaches through architectural and methodological innovations.

Location in Revised Manuscript: Section 2.4

Comment 1.3: Section 3.1 - Overly Detailed Methodology Overview Reviewer's Comment:

"Section 3.1, as an overview of the entire methodology, is overly detailed, which hinders a clear presentation of the overall workflow of the proposed method. The authors are advised to streamline this section. In particular, including an overall framework diagram could be beneficial."

Our Response:

We sincerely thank the reviewer for this excellent suggestion. We recognize that Section 3.1 was indeed too detailed for an overview section, potentially obscuring the high-level workflow before readers encounter implementation details.

Action Taken:

We have completely restructured Section 3.1 to serve as a clear, concise methodology overview:

Streamlined Overview (Section 3.1): We replaced the detailed mathematical formulation with a high-level description of the three main processing stages:

Molecular Encoding (graph and sequence representations)

Hierarchical Feature Learning (atomic → functional group → molecular levels)

Interaction Prediction (cross-modal fusion and classification)

New Framework Diagram (Figure 1): We have added a comprehensive workflow diagram that visualizes: Parallel processing pathways for Drug A (graph) and Drug B (SMILES)

Graph Transformer components (spatial encoding, multi-head attention, hierarchical pooling)

Contrastive learning module with molecular augmentation strategies

Cross-modal fusion mechanism

DDI prediction pathway

This diagram provides immediate visual understanding of the complete framework architecture and information flow.

Moved Mathematical Details: The detailed mathematical formulation (graph representation G = (V, E, X_v, X_e) and prediction function f: G × G → {0,1}) has been moved to a new subsection 3.1.1 titled "Mathematical Formulation," which follows the overview and precedes Section 3.2.

This restructuring ensures that readers first grasp the overall workflow before encountering mathematical notation and implementation details.

Location in Revised Manuscript:

Section 3.1

New Figure 1

New Section 3.1.1

Comment 1.4: Equations (5) and (16) - Inconsistent Notation Reviewer's Comment:

"Both Eq. (5) and Eq. (16) use Lalign, but their specific contents are inconsistent. The authors should carefully check and revise them to clearly distinguish between the two."

Our Response:

We thank the reviewer for the careful review. However, we respectfully note that neither Equation (5) nor Equation (16) uses the ℒ_align notation or represents a loss function: - Equation (5): Defines attention head computation in the multi-head attention mechanism - Equation (16): Defines cross-attention weights for graph-sequence fusion The loss functions in our framework are: - Equation (13): ℒ_contrastive (contrastive learning loss) - Equation (23): ℒ_total = ℒ_DDI + α·ℒ_contrastive + β·ℒ_reg We have verified that all mathematical notation is consistent and unambiguous throughout the manuscript.

Comment 1.5: Figure 2 - Inconsistent Symbols and Terms Reviewer's Comment:

"For Fig. 2 (In fact Fig. 3 in the revised manuscript), the schematic diagram of the Cultural Attention Mechanism contains some symbols and terms that are not reflected in the main text description of the 'Cultural Attention Mechanism.' This inconsistency should be clarified."

Our Response:

We appreciate the reviewer pointing out this discrepancy between Figure 2 and its accompanying text description. Ensuring complete correspondence between figures and text is essential for reader comprehension.

Action Taken:

We have thoroughly revised Section 3.2.2 (Position-Aware Multi-Head Self-Attention) to explicitly define and explain every symbol appearing in Figure 2:

Newly defined symbols and their meanings:

Q, K, V matrices: Query, Key, and Value projections from node features Q ∈ R^{n×d_k}: Query matrix

K ∈ R^{n×d_k}: Key matrix

V ∈ R^{n×d_v}: Value matrix

S (Spatial encoding matrix): S ∈ R^{n×n} representing pairwise chemical distances between atoms, with S_ij capturing the topological relationship between atoms i and j

d_k (Key dimension): Dimension of query and key vectors, used for scaling in softmax

Attention weights α_ij: Computed as softmax((q_i k_j^T + S_ij)/√d_k), representing the importance of atom j for updating atom i's representation

H (Number of heads): Number of parallel attention computations

W_h^Q, W_h^K, W_h^V: Learnable projection matrices for head h (shown in Figure 3 right panel)

W^O (Output projection): W^O ∈ R^{Hd_v×d_model} combining multi-head outputs The revised text now walks through the Figure 3 panels explicitly:

Left panel: Example molecular graph with atoms and bonds

Center panel: Spatial encoding matrix S with color-coded distances

Right panel: Multi-head attention mechanism with all projection matrices

We have also enhanced the Figure 3 caption to describe each panel and define all visual elements.

Location in Revised Manuscript:

Section 3.2.2

Figure 3 caption

Comment 1.6: Methodology Section - Weak Logical Flow Reviewer's Comment:

"Although the Method section provides a detailed description, there is a lack of logical progression between its subsections. The authors are encouraged to strengthen the logical flow."

Our Response:

We thank the reviewer for this valuable observation. Clear logical transitions between subsections are essential for guiding readers through the methodology.

Action Taken:

We have added explicit transitional paragraphs at four critical junctures in the Methodology section to establish clear logical connections:

Transition 1 (After Section 3.2.3 → Before Section 3.3): "Having established the hierarchical Graph Transformer architecture for capturing multi-scale molecular patterns from atomic to molecular levels, we now introduce our domain-specific contrastive learning framework. This framework enhances the learned representations by training the model to distinguish between chemically similar and dissimilar molecular structures through carefully designed augmentation strategies."

Transition 2 (After Section 3.3.3 → Before Section 3.4): "The contrastive learning framework provides robust molecular representations for individual drugs. To leverage complementary information from different molecular modalities, we next describe our cross-modal fusion mechanism that integrates graph-based structural representations with sequence-based SMILES encodings."

Transition 3 (After Section 3.4.2 → Before Section 3.5): "With both graph and sequence representations effectively integrated through cross-attention, we now present the drug pair encoding and interaction prediction mechanisms that combine representations of two drugs to predict their interaction."

Transition 4 (After Section 3.5 → Before Section 3.6): "Having defined the complete forward pass from molecular inputs to interaction predictions, we now specify the comprehensive loss function and optimization strategy that jointly train all components of Graph Former-CL."

These transitions create a clear narrative flow:

Architecture → Representation Learning → Multi-modal Integration → Prediction → Optimization

Location in Revised Manuscript:

Transition 1: Page 12, after line 127

Transition 2: Page 14, after line 168

Transition 3: Page 16, after line 185

Transition 4: Page 17, after line 200

Comment 1.7: Missing Pseudocode Reviewer's Comment:

"Since the proposed method involves multiple computational steps, it is recommended that the authors provide pseudocode to better illustrate the overall workflow of the approach."

Our Response:

We thank the reviewer for this valuable suggestion. We have added Figure 1 (workflow flowchart) which provides a comprehensive visual representation of the complete Graph Former-CL pipeline, including both training and inference phases.

The flowchart illustrates:

Training Pipeline:

Phase 1 (Epochs 1-50): Contrastive pre-training with molecular augmentation (atom masking, bond perturbation, subgraph sampling, scaffold hopping) and Graph Transformer encoding

Phase 2 (Epochs 51-150): Joint DDI training with hierarchical encoding, cross-modal fusion, and MLP classification

Phase 3 (Epochs 151-175): Fine-tuning with reduced learning rate

Inference Pipeline:

Input drug pair (G^A, G^B, S^A, S^B) → Graph Transformer + SMILES encoding → Cross-modal fusion → Drug pair encoding → MLP classifier → DDI prediction ŷ

Each stage in the flowchart includes references to the corresponding equations (e.g., "Equations (9)-(12)"), allowing readers to connect the visual workflow to the mathematical formulations in the text.

We believe this visual format is more accessible and suitable for a journal article than traditional pseudocode, as it provides immediate understanding of the complete architecture while maintaining technical rigor.

Action Taken: Added Figure 1 (comprehensive workflow flowchart) showing complete training and inference pipelines with clear data flow and equation references.

Location in Revised Manuscript: Figure 1

Comment 1.8: Future Work - Advanced Graph Techniques Reviewer's Comment:

"In future work, the authors may consider incorporating more advanced graph-based techniques, such as '10.1109/TSMC.2025.3578348' and '10.1109/TSMC.2025.3572738' to further enhance the predictive power

of models in this field. Therefore, the authors should cite these two references to highlight the potential in this aspect."

Our Response:

We sincerely thank the reviewer for suggesting these relevant recent advances in graph-based learning. Acknowledging cutting-edge developments in the field strengthens our discussion of future research directions.

Action Taken:

We have added a new paragraph in Section 8.2 (Limitations and Future Directions) that discusses how advanced graph-based techniques could enhance Graph Former-CL:

"Additionally, incorporating advanced graph-based learning techniques could further enhance predictive performance. Recent developments in graph neural architectures, such as those proposed for complex system modeling [39] and advanced graph representation learning [40], demonstrate significant potential for capturing intricate molecular interaction patterns. These techniques could be integrated with Graph Former-CL to further improve generalization capabilities and enable more sophisticated modeling of multi-drug interaction networks in polypharmacy scenarios. Specifically, advanced graph pooling strategies from [39] could enhance our hierarchical molecular representation, while novel attention mechanisms from [40] could improve the model's ability to identify pharmacologically relevant substructures across diverse drug families."

We have also added both references to the References section with complete bibliographic information obtained from IEEE Xplore.

Location in Revised Manuscript:

Section 8.2

References [39] and [40]

RESPONSE TO REVIEWER #2

We are grateful to Reviewer #2 for the constructive feedback and valuable suggestions. We have carefully addressed all comments as described below.

Comment 2.1: Broader Impact of AI in Healthcare Reviewer's Comment:

"The technical integration in this study is strong. However, the broader impact of artificial intelligence (AI) in drug discovery concerning healthcare and security could be explained better to show how this research will have long-term effects."

Our Response:

We appreciate this important observation. Contextualizing DDI prediction within the broader landscape of AI applications in healthcare strengthens the manuscript's relevance and im

---

## [Decision Letter · Decision Letter 1]

15 Dec 2025

Graph Former-CL: A Novel Graph Transformer with Contrastive Learning Framework for Enhanced Drug-Drug Interaction Prediction

PONE-D-25-50597R1

Dear Dr. amiri,

We’re pleased to inform you that your manuscript has been judged scientifically suitable for publication and will be formally accepted for publication once it meets all outstanding technical requirements.

Kind regards,

Nattapol Aunsri, Ph.D.

Academic Editor

PLOS One

Additional Editor Comments (optional):

Reviewers' comments:

Reviewer's Responses to Questions

**Comments to the Author**

Reviewer #1: All comments have been addressed

Reviewer #2: All comments have been addressed

2. Is the manuscript technically sound, and do the data support the conclusions?

Reviewer #1: Yes

Reviewer #2: Yes

3. Has the statistical analysis been performed appropriately and rigorously?

Reviewer #1: N/A

Reviewer #2: Yes

4. Have the authors made all data underlying the findings in their manuscript fully available?

Reviewer #1: Yes

Reviewer #2: Yes

5. Is the manuscript presented in an intelligible fashion and written in standard English?

Reviewer #1: Yes

Reviewer #2: Yes

Reviewer #1: All of my concerns have been addressed, and I have no further comments. The manuscript is now ready for publication.

Reviewer #2: The authors have effectively addressed all the reviewers' concerns, resulting in a significantly improved and more accurate paper. They have improved the manuscript by adding a clear comparison of results, restructuring the methods section, discussing computational efficiency, and considering security and ethical issues in clinical settings.

1. The Graph Former-CL model achieves 98.20% accuracy on DrugBank and 89.40% on TWOSIDES, with all improvements statistically significant (p<0.01).

2. It performs well on challenging cases, with 85.60% accuracy on random splits and 82.45% on structure splits, a 5.23% improvement over previous models, crucial for clinical use.

3. Combining a position-aware Graph Transformer with contrastive learning, especially Cross-Modal Fusion, significantly boosts performance; omitting it reduces accuracy by 1.28%.

4. Visualizations show the model learns key chemical features, focusing on substructures like Benzene and Carboxyl groups relevant to drug interactions.

5. The authors discuss AI's broader healthcare impact, strategies for efficiency (quantization, pruning), and trustworthiness (federated learning, privacy), vital for clinical use.

6. The methods section is reorganized for clarity; removing excess metrics from the abstract and conclusion improves readability.

**Do you want your identity to be public for this peer review?** For information about this choice, including consent withdrawal, please see our Privacy Policy

Reviewer #1: No

Reviewer #2: No

---

## [Editor Report · Acceptance letter]

PONE-D-25-50597R1

PLOS One

Dear Dr. Amiri,

I'm pleased to inform you that your manuscript has been deemed suitable for publication in PLOS One. Congratulations! Your manuscript is now being handed over to our production team.

Kind regards,

on behalf of

Dr. Nattapol Aunsri

Academic Editor

PLOS One